# A mechanism for the extension and unfolding of parallel telomeric G-quadruplexes by human telomerase at single-molecule resolution

Bishnu P Paudel[1,2†], Aaron Lavel Moye[3†‡], Hala Abou Assi[4], Roberto El-Khoury[4], Scott B Cohen[3], Jessica K Holien[5], Monica L Birrento[1,2], Siritron Samosorn[6], Kamthorn Intharapichai[7], Christopher G Tomlinson[3], Marie-Paule Teulade-Fichou[8,9], Carlos González[10], Jennifer L Beck[1,2], Masad J Damha[4], Antoine M van Oijen[1,2]*, Tracy M Bryan[3]*

[1]Molecular Horizons and School of Chemistry and Molecular Bioscience, University of Wollongong, Wollongong, Australia; [2]Illawara Health and Medical Research Institute, Wollongong, Australia; [3]Children's Medical Research Institute, University of Sydney, Westmead, Australia; [4]Department of Chemistry, McGill University, Montreal, Canada; [5]School of Science, College of Science, Engineering and Health, RMIT University, Melbourne, Australia; [6]Department of Chemistry and Center of Excellence for Innovation in Chemistry, Faculty of Science, Srinakharinwirot University, Bangkok, Thailand; [7]Department of Biobased Materials Science, Kyoto Institute of Technology, Matsugasaki, Sakyo-ku, Kyoto, Japan; [8]Institut Curie, PSL Research University, Orsay, France; [9]Université Paris Sud, Université Paris-Saclay, Orsay, France; [10]Instituto de Química Física 'Rocasolano', CSIC, Madrid, Spain

*For correspondence:
vanoijen@uow.edu.au (AMO);
tbryan@cmri.org.au (TMB)

†These authors contributed equally to this work

Present address: ‡Stem Cell Program and Divisions of Hematology/Oncology and Pulmonary & Respiratory Diseases, Boston Children's Hospital, Harvard Stem Cell Institute, and Department of Genetics, Harvard Medical School, Boston, United States

Competing interests: The authors declare that no competing interests exist.

**Abstract** Telomeric G-quadruplexes (G4) were long believed to form a protective structure at telomeres, preventing their extension by the ribonucleoprotein telomerase. Contrary to this belief, we have previously demonstrated that parallel-stranded conformations of telomeric G4 can be extended by human and ciliate telomerase. However, a mechanistic understanding of the interaction of telomerase with structured DNA remained elusive. Here, we use single-molecule fluorescence resonance energy transfer (smFRET) microscopy and bulk-phase enzymology to propose a mechanism for the resolution and extension of parallel G4 by telomerase. Binding is initiated by the RNA template of telomerase interacting with the G-quadruplex; nucleotide addition then proceeds to the end of the RNA template. It is only through the large conformational change of translocation following synthesis that the G-quadruplex structure is completely unfolded to a linear product. Surprisingly, parallel G4 stabilization with either small molecule ligands or by chemical modification does not always inhibit G4 unfolding and extension by telomerase. These data reveal that telomerase is a parallel G-quadruplex resolvase.

## Introduction

Human chromosomes contain many guanine (G)-rich elements capable of forming four-stranded G-quadruplex (G4) structures (*Bochman et al., 2012*; *Huppert and Balasubramanian, 2005*; *Maizels and Gray, 2013*; *Rhodes and Lipps, 2015*). A planar G-quartet is formed when four Gs form hydrogen bonds in a cyclical manner via their Hoogsteen faces; such G-quartets stack on top of each other and form G-quadruplexes (*Sen and Gilbert, 1988*; *Sundquist and Klug, 1989*;

*Williamson et al., 1989*). G-rich sequences are primarily located in promoter regions, intron and exon boundaries, origins of replication and telomeres (*Bochman et al., 2012*; *Huppert and Balasubramanian, 2005*; *Maizels and Gray, 2013*; *Rhodes and Lipps, 2015*). Vertebrate telomeres consist of many tandem repeats of the sequence TTAGGG, comprising a double-stranded region and a 3' G-rich overhang (*Blackburn, 1991*). Fluorescence microscopy studies have shown the existence of G4 structures at the telomeric regions of fixed human cells using G4-specific antibodies (*Biffi et al., 2013*; *Liu et al., 2016*). In vitro, telomeric DNA can adopt many different conformations of G4, including inter- or intramolecular forms, arranged in parallel or anti-parallel orientation depending upon the directionality of the DNA backbone (*Burge et al., 2006*). Conditions that may mimic those in the nucleus, including high DNA concentration and water depletion, favour parallel conformations of human telomeric G4 (*Heddi and Phan, 2011*; *Miller et al., 2010*; *Renciuk et al., 2009*). However, the in vivo conformation(s) and biological significance of telomeric G4 in human cells remain elusive and undetermined.

Telomere shortening occurs with every cell division in normal human somatic cells, since conventional DNA polymerases cannot completely synthesize the telomeric end (*Harley et al., 1990*; *Olovnikov, 1971*; *Watson, 1972*). Telomerase, a telomere-specific ribonucleoprotein enzyme complex, extends telomeric DNA in cancer cells, stem cells and cells of the germline, using a unique mechanism of processive rounds of reverse transcription (*Greider and Blackburn, 1985*; *Hiyama and Hiyama, 2007*; *Kim et al., 1994*; *Morin, 1989*; *Podlevsky and Chen, 2012*). Human telomerase minimally consists of the highly conserved telomerase reverse transcriptase protein (hTERT) and an RNA component containing an 11 nucleotide (nt) template sequence (hTR) (*Wyatt et al., 2010*). Telomerase extends its telomeric substrate by first binding to telomeric DNA, followed by hybridization of the RNA template with the DNA. Second, telomerase extends the substrate DNA by using its RNA as a template for nucleotide addition to the DNA 3' end. Once the template 5' boundary has been reached, telomerase translocates downstream, resulting in re-alignment of the RNA template with the new 3' end of the product DNA (*Greider and Blackburn, 1989*; *Podlevsky and Chen, 2012*).

One function of telomeric G4 may be to act as a 'cap', protecting the telomere from DNA degradation (*Smith et al., 2011*). It has been widely believed that G4 formation within the 3' telomeric overhang blocks telomere extension by telomerase, since early in vitro studies had shown that G4 structures inhibit telomerase extension of a telomeric DNA substrate (*Zahler et al., 1991*; *Zaug et al., 2005*). In addition, stabilization of G4 with small-molecule ligands has been shown to more effectively inhibit telomerase activity, suggesting that chemical stabilization of G4 structures may be a viable anti-cancer therapeutic strategy (*De Cian et al., 2007a*; *Gomez et al., 2016*; *Sun et al., 1997*). However, the above-mentioned studies did not distinguish between G4 conformations, and used oligonucleotides that likely folded into anti-parallel or 'hybrid' G4 forms as a telomerase substrate. A variety of helicases, such as RECQ5, WRN, BLM, FANC-J and RHAU, recognize and resolve G4 in a conformation-specific manner (*Budhathoki et al., 2016*; *Liu et al., 2010*; *London et al., 2008*; *Tippana et al., 2016*). Similarly, we have demonstrated that telomerase can bind and extend specific conformations of telomeric G4 - those that are parallel-stranded and intermolecular - and that this property of telomerase is well conserved from ciliates to human (*Moye et al., 2015*; *Oganesian et al., 2007*; *Oganesian et al., 2006*). This suggests that if telomeric G-quadruplexes were to adopt a parallel conformation in vivo, they may indeed be extended by telomerase, contrary to what had been previously hypothesized.

Telomerase extension reactions in the presence of individual nucleotides demonstrated that the 3' end of a G4 substrate aligns correctly with the RNA template of telomerase, suggesting that at least partial resolution of G4 structure is required for telomerase extension (*Moye et al., 2015*). However, the evidence for partial disruption of the G4 was indirect, and a mechanistic understanding of telomerase resolution of G4 DNA, in the absence of a known helicase function of hTERT, remained elusive. Furthermore, since all previously-tested substrates were both parallel-stranded and intermolecular, it was unknown which of these properties form the basis of telomerase recognition. To directly study the mechanistic details of parallel G4 unfolding and extension by human telomerase, here we combined ensemble telomerase enzymatic assays and single-molecule FRET (smFRET) measurements in vitro, using both intramolecular and intermolecular parallel G4 as FRET sensors. We provide direct evidence that wild-type human telomerase can unfold both intra- and intermolecular parallel G4 completely in the presence of dNTPs, using a 3-step mechanism. First,

telomerase binds the G4, partially changing its conformation in a process that involves binding of the hTR template sequence. Second, telomerase adds individual nucleotides to the 3' end of the partially unfolded parallel G4. Lastly, the translocation of telomerase results in complete disruption of the G4 structure. Unexpectedly, stabilization of parallel intermolecular G4 using different G4 ligands did not inhibit telomerase unfolding and extension of this substrate. Overall, we provide a mechanistic explanation of conformation-specific telomerase extension of telomeric G4 at single-molecule resolution and demonstrate that small molecule-mediated inhibition of telomerase extension of telomeric G4 is topology-dependent.

## Results

### Human telomerase binds, unfolds and extends intramolecular parallel G-quadruplexes

Parallel, intermolecular G4 are substrates for telomerase, whereas intramolecular antiparallel or hybrid conformations are not (*Hwang et al., 2014*; *Lee et al., 2017*; *Moye et al., 2015*; *Oganesian et al., 2006*; *Zahler et al., 1991*; *Zaug et al., 2005*). Determining whether it is the parallel or intermolecular nature of G4 structures that allows their recognition by telomerase has been difficult, since a 4-repeat human telomeric oligonucleotide does not readily fold into stable parallel intramolecular G4 at the concentrations used in in vitro assays, and instead exists as a mixture of topologies under most conditions (*Dai et al., 2007*; *Long and Stone, 2013*; *Palacký et al., 2013*; *Petraccone et al., 2012*). For this reason, we made use of the modified nucleotide 2'-fluoro-arabino-guanosine (2'F-araG), which induces parallel propeller-type G4 conformations (*Peng and Damha, 2007*). We have previously demonstrated that substitution of six guanosines in the telomeric sequence AGGG(TTAGGG)$_3$ with 2'F-araG leads to a 15°C increase in $T_m$ of the resulting intramolecular G4, and a shift from the usual antiparallel or hybrid topology of this sequence in potassium solution (*Lim et al., 2009*; *Renciuk et al., 2009*) to a parallel conformation (*Figure 1A*; *Abou Assi et al., 2017*). Here, we demonstrate that G4 formed from the unmodified sequence (22G0) in KCl is a poor telomerase substrate, leading to the previously-observed stuttering pattern (*Zaug et al., 2005*) in a direct telomerase extension assay involving incorporation of radiolabeled $\alpha^{32}$P-dGTP (*Figure 1B*, lane 3). Substitution with 2'F-araG (22G3; see *Supplementary file 1* for sequence) restored the expected 6-nt repeat pattern of telomerase extension (*Figure 1B*, lane 2), despite the increase in thermal stability of this G4. Thus, telomerase is able to extend parallel G4 structures, whether they are inter- or intramolecular.

To determine the mechanism of extension of intramolecular G4, we designed a version of 22G3 with a FRET donor dye (AlexaFluor 555) on one of the propeller loops, and an extended 5' tail; circular dichroism (CD) spectroscopy demonstrated that the G4 formed from this sequence retained a predominantly parallel topology (*Supplementary file 1* and *Figure 1C and D*; oligonucleotide 22G3 +tail). The extended 5' tail enabled hybridization with a second DNA oligonucleotide containing a FRET acceptor dye (AlexaFluor 647) and a biotin for surface immobilization in smFRET studies; we refer to the assembled FRET-modified G4 as F-22G3 (*Figure 1C*). F-22G3 also retained a parallel G4 topology, with a slight shoulder at ~290 nm attributable to the duplex portion of the molecule (*Figure 1D*). Assembled F-22G3 was efficiently extended by telomerase, and was possibly an even better substrate than 22G3, with the expected 6-nt repeat pattern (*Figure 1B*, lane 1).

We predicted that this structure would yield a high FRET ratio as the donor and acceptor fluorophores would be in close proximity (*Figure 1C*), but when unfolded the strands would move apart and a low FRET ratio would be expected. We carried out smFRET experiments using the modified F-22G3 construct in KCl by tethering it on pegylated coverslips via a biotin-streptavidin-biotin linkage. Surface-immobilized G4s were excited using a 532 nm laser and signals emitted from both the donor and acceptor fluorophores were collected and their intensities measured over time (*Figure 2A*, top panel); both dyes provided a constant fluorescence signal over several minutes. The apparent FRET values between these two fluorophores were calculated by dividing the acceptor intensity by the sum of the donor and acceptor intensities (*Figure 2A*, bottom panel). Individual F-22G3 molecules all displayed a constant FRET ratio of ~0.5 over time, indicating that they did not undergo any detectable conformational changes in the experimental time window. Given a Förster radius between AF555 and AF647 of 51 Å, the calculated FRET efficiency for this distance is 0.44,

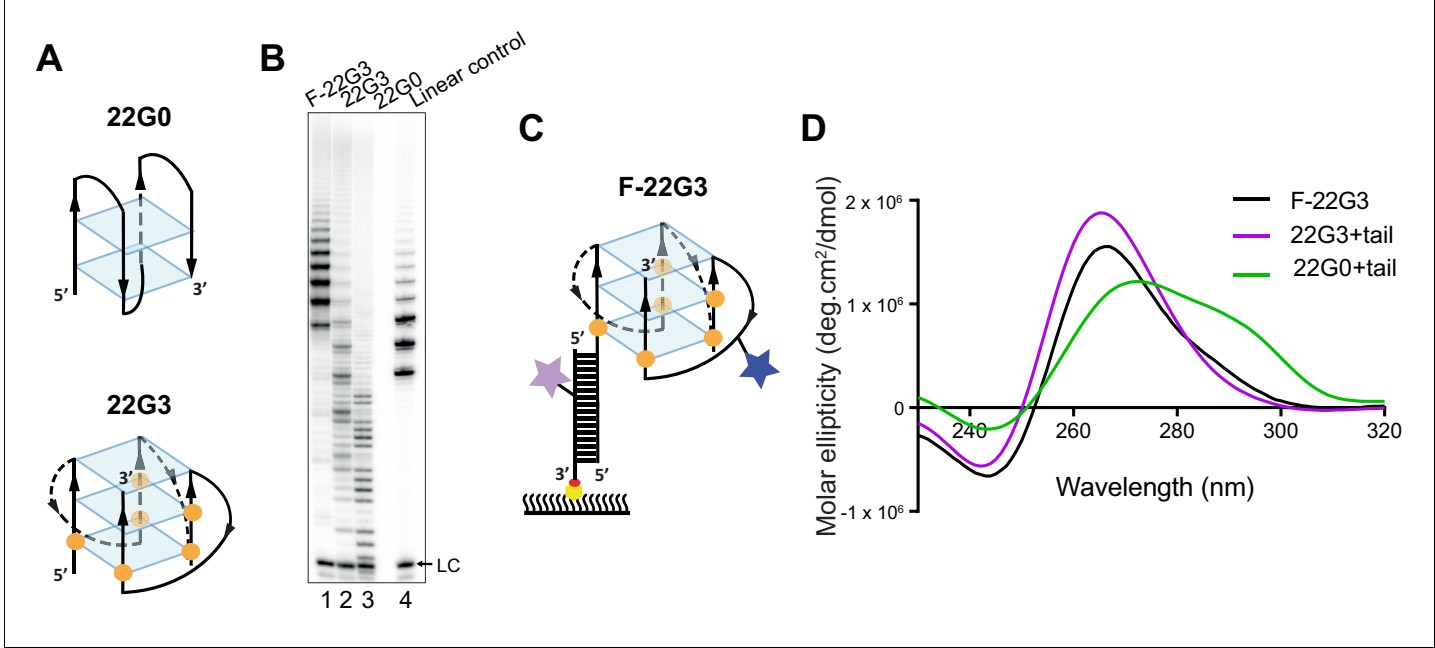

**Figure 1.** Telomerase extends a parallel intramolecular G-quadruplex. (**A**) Schematic of the likely topology of (**top**) an antiparallel G-quadruplex formed from unmodified telomeric 22-mer 22G0 and (bottom) a parallel unimolecular G-quadruplex formed from a telomeric 22-mer with 2′F-araG in the indicated positions (orange circles). (**B**) Telomerase extension assays in the presence of radiolabeled dGTP ($\alpha$[32]P-dGTP). The extension products of 2 µM each of fluorescently-labeled 22G3 (F-22G3), unlabeled 22G3, unmodified 22G0 and linear control oligonucleotide Bio-L-18GGG (see *Supplementary file 1*) were analyzed using denaturing polyacrylamide gel electrophoresis. LC: 5′-[32]P-labeled synthetic 12 nt DNA used as a recovery/ loading control. (**C**) The F-22G3 FRET construct, showing the positions of the FRET dye pair and biotin for immobilization onto the functionalized coverslip. Blue star: AlexaFluor 555 (donor dye); purple star: AlexaFluor 647 (acceptor dye); orange circles: 2′F-araG; red circle: biotin; yellow circle: Neutravidin. (**D**) CD spectra of G4 (20 µM) formed from 2′F-araG-modified 5′-extended 22G3 labeled with a FRET dye (22G3+tail), and the same G4 after hybridization to a second oligonucleotide bearing a second FRET dye (F-22G3). A peak at 265 nm is characteristic of parallel G4; the slight shoulder at ~290 nm in the F-22G3 spectrum is attributable to duplex DNA since it is absent from the spectrum of 22G3+tail. 22G0+tail is a control oligonucleotide of equal length and base composition to 22G3+tail, but with no 2′F-ara G substitutions or conjugated dye.

which is close to our experimental FRET value of 0.5. To confirm that this FRET value represents a G4 structure, we substituted the KCl-containing buffer for a LiCl solution, since Li[+] does not support G4 formation; in a majority of molecules, the FRET signal decreased from ~0.5 to~0.14 and remained at the low FRET state during the entire observation window (*Figure 2—figure supplement 1*). Using ~100 smFRET trajectories pooled from multiple independent experiments in KCl (sample sizes for each condition indicated in figure panels), a FRET heat map was constructed, showing the distribution of average FRET values (*Figure 2B*). The heat map showed that the mean FRET value (0.53 ± 0.05) remained unchanged over time; an alternative histogram representation of the same data grouped into 15 s bins confirms this conclusion (*Figure 2—figure supplement 2A*). These data demonstrate that the F-22G3 G-quadruplex is stable over time.

Next, we tested whether telomerase presence affects the F-22G3 structure. To this end, we imaged F-22G3 in the presence of catalytically active telomerase, but in the absence of deoxynucleotide triphosphates (dNTPs). Approximately 65% of F-22G3 molecules showed an abrupt drop in FRET value, from 0.53 ± 0.05 to 0.3 ± 0.1, during the 160 s after telomerase was injected into the microscopic channel containing immobilized F-22G3 (*Figure 2C and D*). The remaining 35% of molecules did not show any change in FRET signal over the observed time; it is possible that the binding reaction had not proceeded to completion within this time period, or that a subpopulation of enzyme or DNA molecules are incompetent for binding. We collected 125 molecules showing a step-wise change in FRET value and plotted the data in a FRET heat map and a histogram plot as a function of time; both plots showed a drop in mean FRET value from ~0.53 to~0.3 FRET over this time (*Figure 2E* and *Figure 2—figure supplement 2B*). We interpret this to represent telomerase

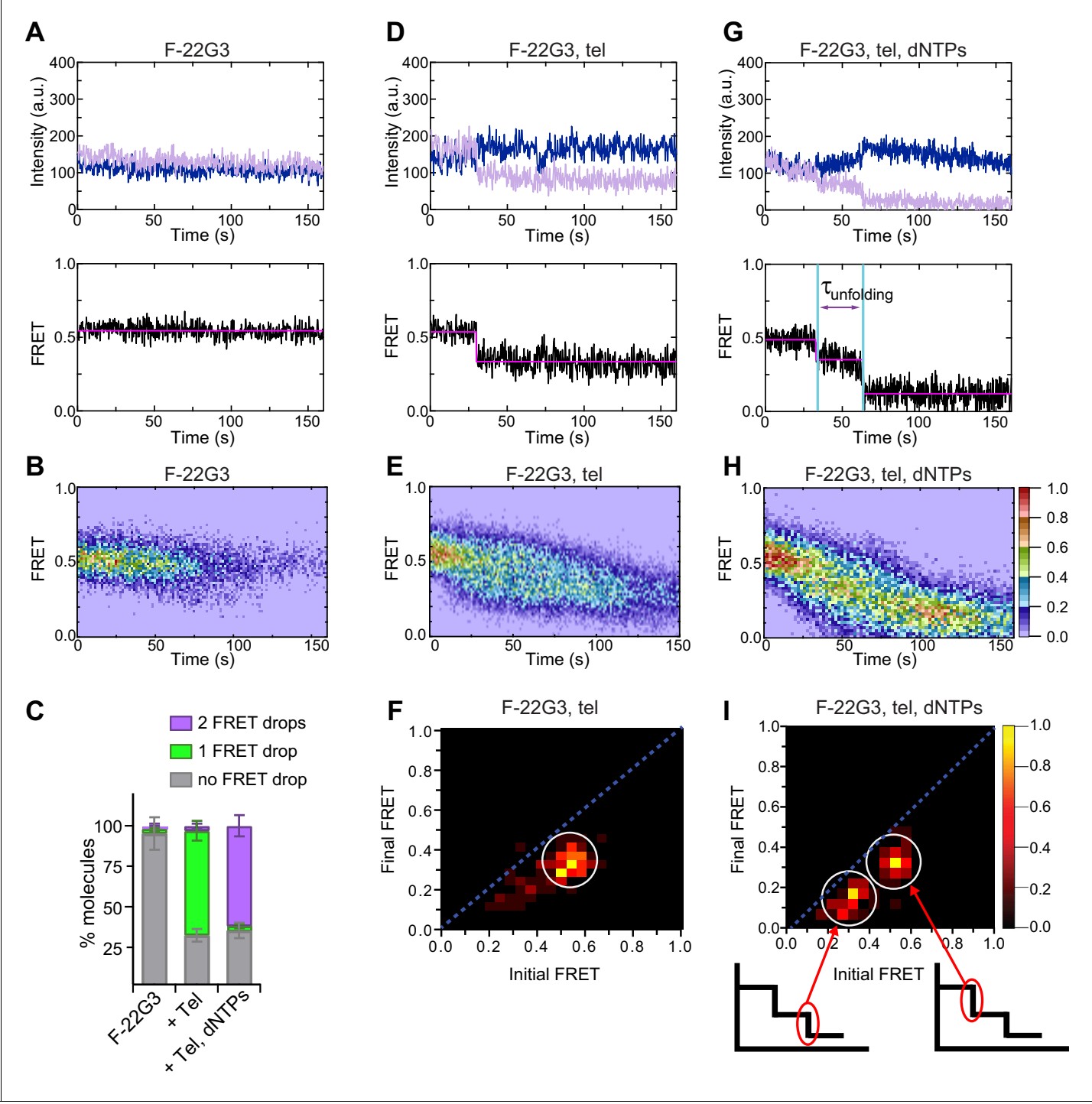

**Figure 2.** Telomerase unfolds a parallel intramolecular G-quadruplex. (**A**), (**D**), (**G**) Representative single-molecule acceptor (purple) and donor (blue) intensities of F-22G3 molecules over time (top panels), and the FRET traces (bottom panels) representing the ratio of acceptor intensity to the sum of acceptor and donor intensities, of either F-22G3 alone (**A**), in the presence of telomerase (**D**), or in the presence of telomerase and dNTPs (**G**). (**B**), (**E**), (**H**) Heat maps of the distribution of FRET intensities over 0–150 s of F-22G3, either alone (n = 99) (**B**), in the presence of telomerase (n = 125) (**E**), or in the presence of telomerase and dNTPs (n = 90) (**H**). All plots include molecules collected in 4–6 independent experiments. For color key, see panel (**H**). (**F**), (**I**) Transition density plots (TDPs) showing the change in FRET value of molecules from experiments in (**E**) and (**H**) (n = 90 and 75, respectively). Schematics in (**I**) show assignment of each TDP peak to one of the two steps of FRET reduction observed in single molecule traces. (**C**) Plot of the percentage of molecules showing no change in FRET, a single FRET drop, or a two-step FRET drop, in the experiments shown in (**A**) to (**I**); error bars represent sample size/√total population.

*Figure 2 continued on next page*

*Figure 2 continued*

The online version of this article includes the following source data and figure supplement(s) for figure 2:

**Source data 1.** source data for the graph shown in *Figure 2C*.
**Figure supplement 1.** Control demonstrating G4 origin of ~0.5 FRET value of F-22G3.
**Figure supplement 2.** Histogram representations of F-22G3 FRET data.
**Figure supplement 3.** Control for effect of dNTPs on F-22G3 FRET.
**Figure supplement 4.** The unfolding rate of F-22G3, calculated by fitting the dwell time distributions of the intermediate FRET state (see *Figure 2G*) to a gamma distribution (n = 76 molecules).

binding to F-22G3 and partially opening the structure, which then remained stable in its new conformation.

We confirmed this conclusion by quantitatively analyzing the FRET changes during the transitions. For all molecules that showed a change in FRET signal over time, the frequency with which molecules transitioned between states was determined using state finding algorithm vbFRET (https://sourceforge.net/projects/vbfret/; *Bronson et al., 2013*). Then, the transition frequencies were plotted as a function of initial and final FRET states to obtain transition density plots (TDP) (*Figure 2F*). In the presence of telomerase, the TDP showed a single cluster of transitions at initial FRET ~ 0.5 and final FRET ~ 0.3, consistent with the shift in mean FRET in the heat map.

To examine changes in F-22G3 structure during its extension by telomerase, we performed smFRET experiments in the presence of both telomerase and dNTPs. Under these conditions, ~65% of molecules showed a two-step drop in FRET values, from 0.53 ± 0.05 to 0.3 ± 0.1, and then to 0.15 ± 0.05 (*Figure 2C and G–I*, and *Figure 2—figure supplement 2C and D*). The FRET decrease from high to low FRET states in these events was irreversible, supported by the presence of two off-diagonal clusters in the TDP (*Figure 2I*), suggesting a continuous irreversible unfolding of G4 structure. As a control, we performed smFRET experiments in the presence of dNTPs alone and observed no change in FRET signal (*Figure 2—figure supplement 3*). These data suggest that telomerase disrupts F-22G3 structure completely in the presence of dNTPs.

The rate of F-22G3 unfolding upon telomerase binding in the presence of dNTPs was measured by measuring the dwell time distribution at the 0.3 FRET state ($\tau_{unfolding}$; *Figure 2G*) and fitting the distribution to a gamma distribution (*Figure 2—figure supplement 4*). F-22G3 molecules exhibited telomerase-mediated unfolding with a rate constant of $k_{unfolding} = 0.120 \pm 0.013$ s$^{-1}$ (mean ± SEM). This unfolding rate is comparable with the rate of unfolding of parallel G4 by Pif1 helicase (0.11 s$^{-1}$) (*Byrd and Raney, 2015*). Gamma distributions also provide information about the number of rate-limiting steps within a complex kinetic process (*Floyd et al., 2010*); in this case, the number of steps (*N*) is 2.8 (*Figure 2—figure supplement 4*), suggesting that unfolding is a complex process involving at least 2–3 kinetic steps.

## Human telomerase also binds, unfolds and extends intermolecular parallel G-quadruplexes

We have previously demonstrated that a tetrameric, parallel G4 (*Gavathiotis and Searle, 2003*) formed from four copies of the 7-mer telomeric sequence TTAGGGT in K$^+$ is highly stable but can be extended by human telomerase (*Moye et al., 2015*). This sequence provides the advantage that the conformation of its G4 topology is unambiguous; the sequence is too short to form intramolecular G4 structures, and can only exist as a parallel tetramer. To examine whether this tetramer is extended by telomerase using the same mechanism as an intramolecular G4, we prepared a version of this quadruplex labeled with a pair of FRET dyes and a biotin with which to immobilize the DNA. Four different strands, each consisting of the sequence TTAGGGT (here called 7GGT) and a hexaethylene glycol spacer, were annealed in an equimolar mixture in KCl solution. Three of these strands were 5' modified with either AlexaFluor 555 (donor dye), AlexaFluor 647 (acceptor dye) or biotin, and the remaining strand was unmodified (*Figure 3A*). We assembled an equimolar mixture of each of these modified oligonucleotides to produce G4s with different combinations of modifications. The resulting mixture of G4s were all parallel-stranded, as confirmed using CD spectroscopy, and had an average melting temperature similar to that of the unmodified G4 ([7GGT]$_4$; *Figure 3—figure supplement 1*). Direct telomerase activity assays demonstrated that human telomerase can

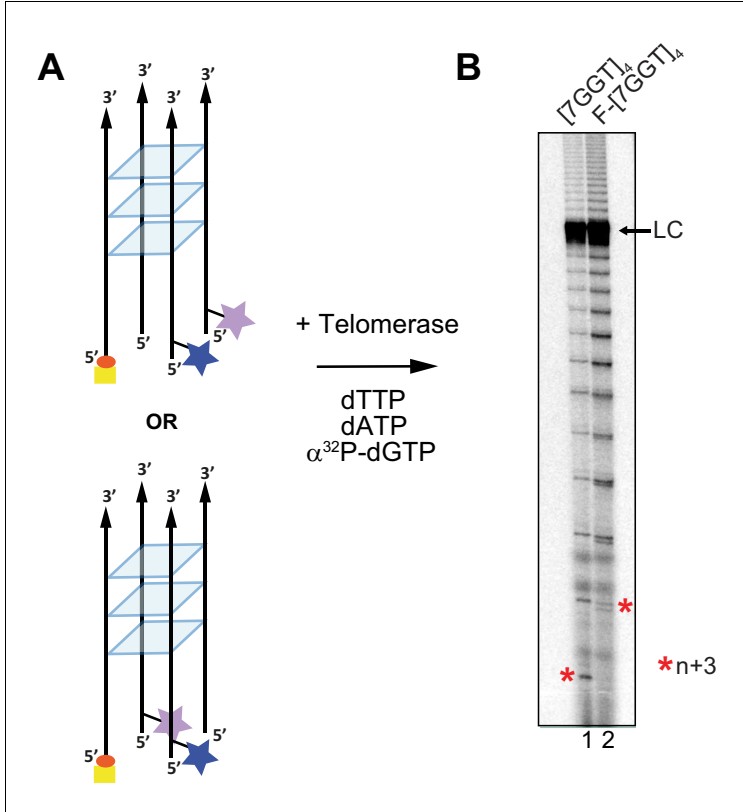

**Figure 3.** Telomerase extends a tetrameric parallel G-quadruplex. (A) Schematic representation of F-[7GGT]$_4$ used in smFRET studies. Blue star: AlexaFluor 555 (donor dye); purple star: AlexaFluor 647 (acceptor dye); red circle: biotin; yellow square: Neutravidin. (B) Telomerase extension assays using either 1 µM of unmodified [7GGT]$_4$ or the version containing the FRET pair dyes (F-[7GGT]$_4$); radiolabeled extension products were electrophoresed on a denaturing polyacrylamide gel (* indicates the position of n+3 products in the gel). LC: 5′-$^{32}$P-labeled synthetic 100 nt DNA used as a recovery/loading control.

The online version of this article includes the following figure supplement(s) for figure 3:

**Figure supplement 1.** Properties of tetrameric G4 with added FRET dyes.

extend the FRET-modified G4 construct (which we refer to as F-[7GGT]$_4$) as efficiently as the unmodified [7GGT]$_4$ (*Figure 3B*). In single-molecule microscopy analyses, we analyzed only G4 structures containing a single copy of each of the four strands, through selection during post-image processing and data analysis (see Methods for details). Note that there are two possible orientations of the positions of the two dyes (on adjacent strands or on diagonally opposite strands, as depicted in *Figure 3A*), but it is unlikely that the distance between the dyes in these two conformations is sufficiently different to resolve by FRET. We note that reliable distance estimates obtained from FRET are extremely challenging, with the FRET efficiency depending upon several factors beyond distance: 1) The length of linkers that are used to attach each fluorescent dye (in our case a C6 carbon linker) will affect dye mobility and increase the distance between dyes (*Sindbert et al., 2011*), 2) the quenching of emission due to the presence of Gs in the dye's vicinity (*Sindbert et al., 2011*), and 3) solvent polarity (*Ishikawa-Ankerhold et al., 2012*). smFRET experiments showed that most F-[7GGT]$_4$ molecules exhibit a steady FRET signal at a ratio of ~0.6, and the FRET signal did not change over time (*Figure 4A and B*, *Figure 4—figure supplement 1A*), indicating formation of a stable parallel intermolecular G4. Substitution of the K$^+$-containing buffer for one containing Li$^+$ resulted in a drop to a stable FRET value of ~0.2, followed by complete loss of the signal from both dyes, likely representing G4 unwinding followed by loss of the non-biotinylated dye-bearing strands (*Figure 4—figure supplement 2*). Telomerase alone was sufficient for partial unwinding of parallel F-[7GGT]$_4$, as demonstrated by a drop in FRET state from ~0.6 to ~0.4 with time (*Figure 4D–F*, *Figure 4—figure supplement 1B*). About 60% of molecules showed this one-step drop in FRET signal

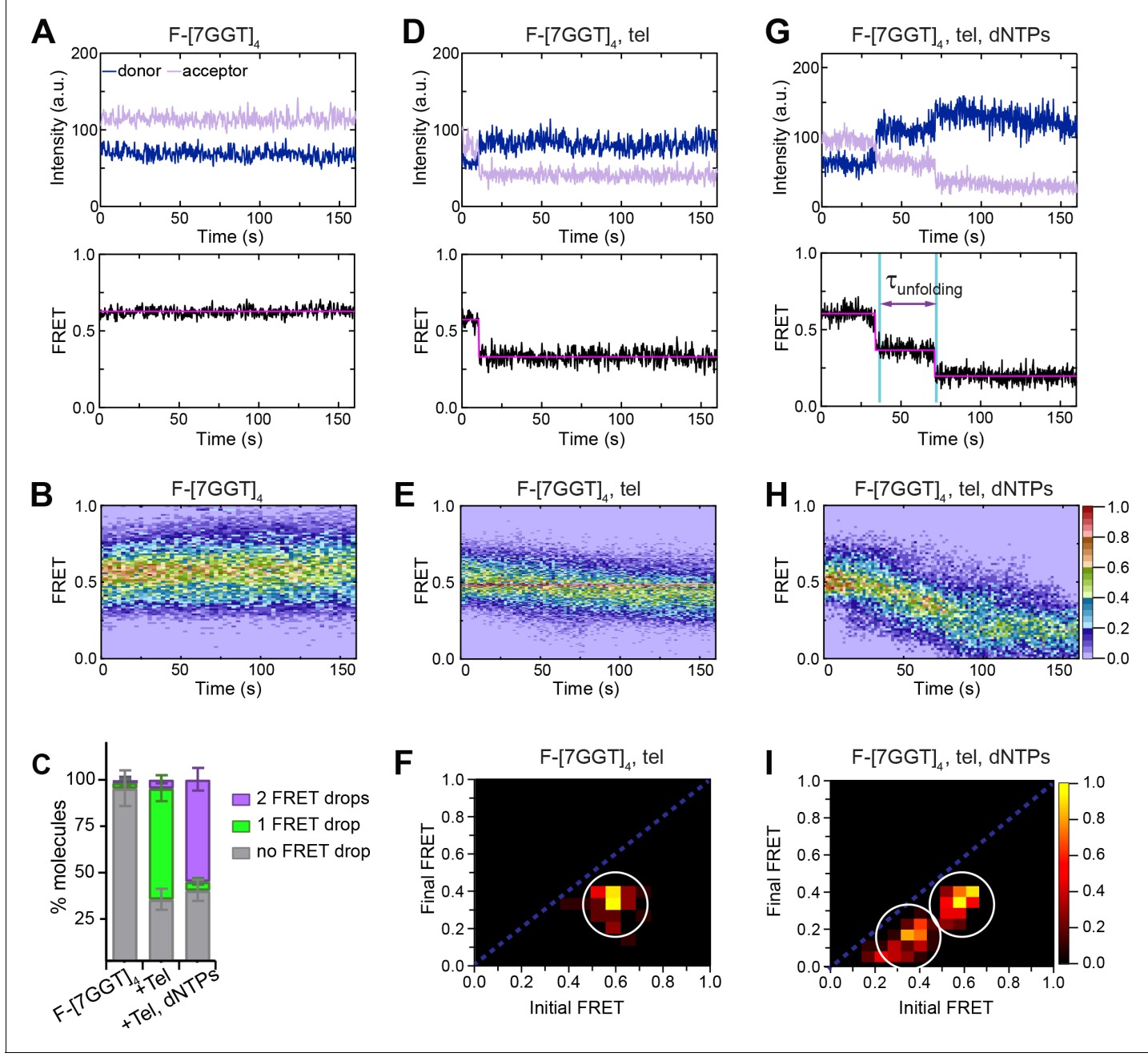

**Figure 4.** Telomerase unfolds a tetrameric parallel G-quadruplex. (A), (D), (G) Representative single-molecule acceptor (purple) and donor (blue) intensities of F-[7GGT]₄ molecules over time (top panels), and the corresponding FRET traces (bottom panels). (B), (E), (H) Heat maps of the distribution of FRET intensities over 0–150 s. For color key, see panel (H). Panels (A) and (B) represent F-[7GGT]₄ alone (n = 105 molecules), (D) and (E) show F-[7GGT]₄ in the presence of telomerase (n = 87), and (G and H) Show F-[7GGT]₄ in the presence of telomerase and dNTPs (n = 81). All plots include molecules collected in 4–6 independent experiments. (C) Plot of the percentage of molecules showing no change in FRET, a single FRET drop, or a two-step FRET drop, in the experiments shown above; error bars represent sample size/√total population. (F), (I) TDPs depicting initial and final FRET states of all molecules that showed a change in FRET value, in the presence of telomerase (n = 65) (F) or telomerase and dNTPs (n = 75) (I). For color key, see panel (I).

The online version of this article includes the following source data and figure supplement(s) for figure 4:

**Source data 1.** source data for the graph shown in *Figure 4C*.

**Figure supplement 1.** Histogram representations of F-[7GGT]₄ data.

**Figure supplement 2.** Control demonstrating G4 origin of ~0.5 FRET value of F-[7GGT]₄.

**Figure supplement 3.** Control for effect of dNTPs on F-[7GGT]₄ FRET.

*Figure 4 continued on next page*

*Figure 4 continued*

**Figure supplement 4.** Unfolding rates of F-[7GGT]$_4$ under different experimental conditions.

upon injection of telomerase (*Figure 4C*). Note that the distribution of initial FRET values of those molecules that show a change in FRET value is not as broad as the FRET distribution of the whole population of molecules (compare the spread at 0 time in *Figure 4E* to that in *Figure 4B*), supporting the interpretation that those G-quadruplexes that do not bind to telomerase have folded improperly or are folding intermediates, and thus show a high level of dynamic behaviour that is too fast to resolve and results in a broadening of the FRET peak. In the case of the 60% of molecules that do show a drop in FRET signal, we interpret this to demonstrate that telomerase results in a partial separation of the FRET dyes upon binding to one strand of the tetrameric G4.

In the presence of telomerase and dNTPs, most F-[7GGT]$_4$ molecules experienced a two-step drop in FRET, from $0.57 \pm 0.05$ to $0.36 \pm 0.04$, and then to $0.19 \pm 0.05$ (*Figure 4C and G–I*, *Figure 4—figure supplement 1C and D*); this transition did not occur in the presence of dNTPs alone (*Figure 4—figure supplement 3*). The two FRET transition clusters observed by TDP analysis (*Figure 4I*) suggest that telomerase unfolds parallel intermolecular G4 irreversibly. The low FRET state of ~0.2 is consistent with the low FRET state observed during Li$^+$-mediated G4 unfolding (*Figure 4—figure supplement 2*); surprisingly, however, we did not observe complete loss of FRET signal (i.e. disappearance of both dyes simultaneously), as we had seen in the presence of Li$^+$. Instead, the majority of molecules exhibited a stable FRET signal of ~0.2, followed by acceptor dye photobleaching then later photobleaching of the donor dye. This suggests that while telomerase extension results in G4 disruption, the presence of the enzyme keeps the four strands in proximity of each other, rather than allowing them to completely diffuse apart.

The rate of unfolding of F-[7GGT]$_4$ by telomerase in the presence of dNTPs was measured by plotting the dwell times of the intermediate transition states (FRET value of 0.4) and fitting them to a gamma distribution that yielded $k_{unfolding} = 0.073 \pm 0.013$ s$^{-1}$ (mean $\pm$ SEM; *Figure 4—figure supplement 4A*), with $N = 2.3 \pm 0.1$. Thus, the rate of unfolding of [7GGT]$_4$ by telomerase in the presence of dNTPs is comparable to that of F-22G3. The gamma distribution also suggested that 2–3 rate-limiting steps contribute to this rate. Since the transitions between FRET states were very abrupt for both G4s (see *Figure 2G* and *Figure 4G*), it was possible that further rate-limiting steps occur during these transitions, which are too fast to resolve. To explore this, we carried out smFRET analyses of [7GGT]$_4$ in the presence of a 100-fold lower concentration of dGTP; this resulted in ~40% of traces that showed a plateau at FRET ~ 0.4 followed by a gradual transition to ~0.2 (example in *Figure 4—figure supplement 4B*). Quantitation of the dwell times at FRET 0.4 showed a $k_{unfolding}$ of $0.040 \pm 0.006$ s$^{-1}$ (mean $\pm$ SEM; *Figure 4—figure supplement 4C*), which is ~1.8 fold slower than the rate at a high dGTP concentration, and $N = 2.10 \pm 0.03$. Approximation of the times taken for the gradual transition to FRET 0.2 (that we refer to as $\tau_{unfolding-2}$) yielded a rate of $0.44 \pm 0.14$ s$^{-1}$, with 3–4 rate-limiting steps revealed by the gamma distribution (*Figure 4—figure supplement 4D*). Together, these data suggest that the process of G4 unfolding by telomerase is complex and involves at least five independent steps, whose rates are influenced by dNTP concentration.

## Complete G-quadruplex unwinding by telomerase requires its catalytic activity

Next, we asked if either of the two step-wise drops in FRET values are dependent upon the nucleotide incorporation activity of telomerase. Synthesis activity requires three conserved aspartate residues in the reverse transcriptase domain of the TERT protein (*Counter et al., 1997*; *Harrington et al., 1997*; *Lingner et al., 1997*; *Weinrich et al., 1997*). Mutation of any one of these aspartates results in loss of telomerase catalytic activity but retention of its ability to bind to a DNA primer (*Wyatt et al., 2007*). We introduced an aspartate-to-alanine mutation at hTERT amino acid 712 and confirmed that this mutant telomerase (D712A) lost all primer extension activity (*Figure 5—figure supplement 1*). smFRET experiments with F-[7GGT]$_4$ demonstrated an initial drop in FRET after addition of D712A telomerase, but no further drop in FRET was observed upon addition of dNTPs (*Figure 5A–E*). This result suggests that binding of telomerase to F-[7GGT]$_4$, resulting in

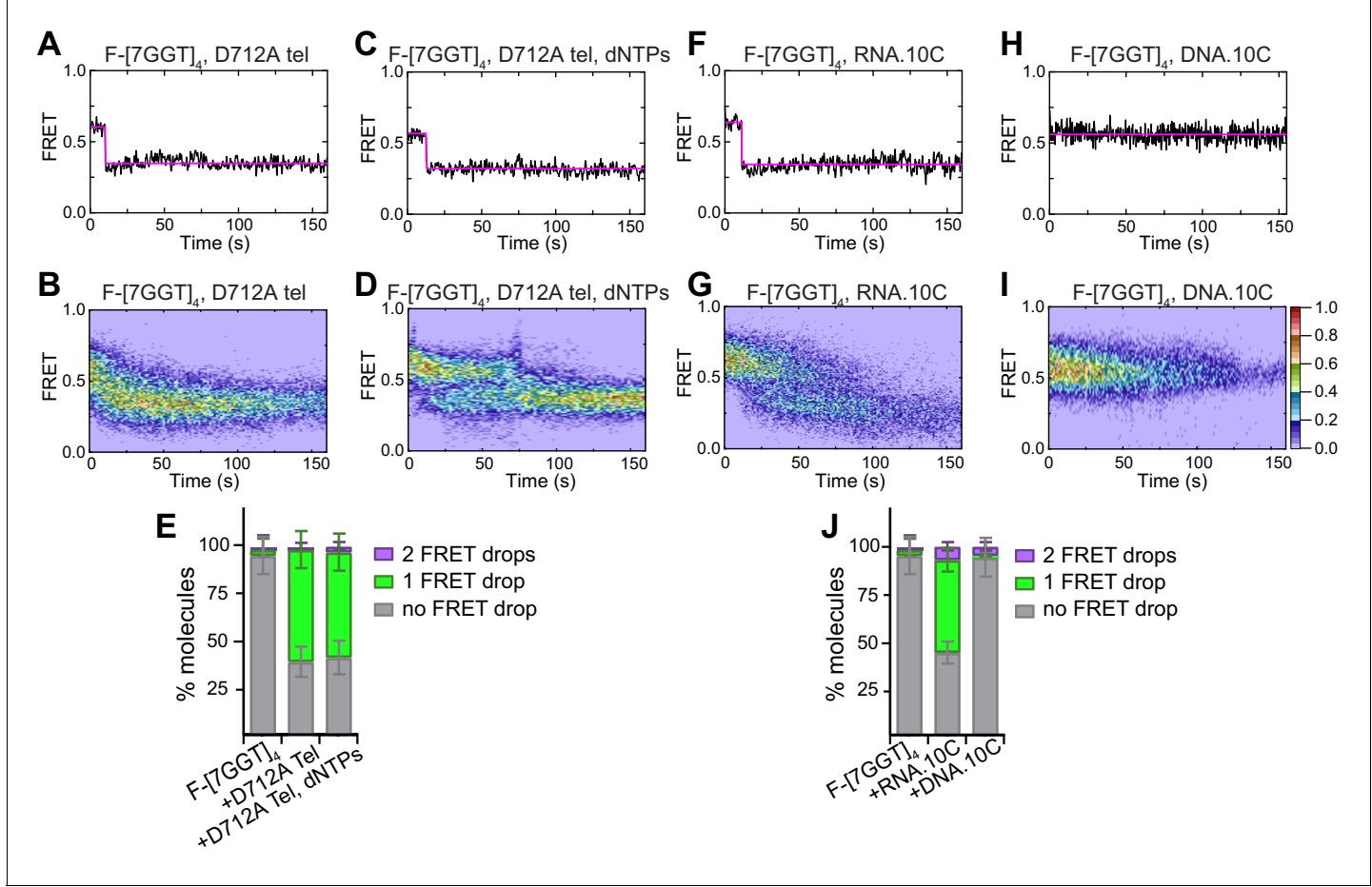

**Figure 5.** Partial unfolding of G4 does not require telomerase catalytic activity, and can be induced by the RNA template. (A), (C), (F), (H) Examples of individual F-[7GGT]$_4$ FRET traces under the indicated experimental conditions over 160 s. (B), (D), (G), (I) Heat maps of the distribution of FRET trajectories over 0–150 s using 80, 75, 95 and 129 molecules, respectively, under the indicated experimental conditions. All plots include molecules collected in 4–6 independent experiments. For color key, see panel (I). (E), (J) Plots of the percentage of molecules showing no change in FRET, a single FRET drop, or a two-step FRET drop, in the experiments shown above; error bars represent sample size/$\sqrt{}$total population.

The online version of this article includes the following source data and figure supplement(s) for figure 5:

**Source data 1.** Source data for the graph shown in *Figure 5E*.
**Source data 2.** Source data for the graph shown in *Figure 5J*.
**Figure supplement 1.** Telomerase activity assay using purified catalytically inactive human telomerase (D712A) compared with wild type (WT) telomerase.
**Figure supplement 2.** Effect of RNA.10C concentration on F-[7GGT]$_4$ FRET.

partial G4 unwinding, is independent of telomerase catalytic activity, but full unwinding of the G4 requires its extension by telomerase.

## The RNA template sequence is involved in partial unfolding of G-quadruplex structure

It is known that the RNA template of telomerase plays an important role in recognizing and binding the telomeric end by canonical base paring. We have previously demonstrated that [7GGT]$_4$ is extended by the nucleotides that would be predicted from canonical base-pairing with the RNA template (*Moye et al., 2015*). Therefore, we hypothesized that the RNA template binds to the G4, facilitating the opening of the structure. To test this, we performed smFRET experiments in the presence of a 10-nt RNA oligonucleotide mimicking the human telomerase template sequence, RNA.10C (*Supplementary file 1*). At a high concentration of RNA.10C (500 µM), a majority of F-[7GGT]$_4$ molecules executed a single-step drop in FRET from 0.57 to 0.36 (*Figure 5F,G and J*);

thus, a short oligonucleotide resembling the telomerase template induces G4 unfolding in a similar manner to the whole telomerase enzyme. The fraction of the G4 population that showed a drop in FRET signal was dependent on the concentration of RNA.10C (*Figure 5—figure supplement 2A and B*), and at both concentrations ~ 20% of the molecules showed dynamic behaviour during the observation window, transiently returning to the high FRET state (examples in *Figure 5—figure supplement 2C and D*), which was not observed in experiments with the whole telomerase enzyme. This indicates that the affinity of the RNA oligonucleotide for the G4 is lower than that of telomerase, which is not unexpected given the multiple protein-DNA contacts that can occur in the presence of hTERT. Nevertheless, the effect of an oligonucleotide on G4 structure was very specific to RNA; the presence of 500 μM of a 10-nt DNA oligonucleotide of identical sequence (DNA.10C) did not stimulate any change in the FRET signal of F-[7GGT]$_4$ over the same time period (*Figure 5H–J*). These data suggest that the hTR template is specifically involved in binding to the G4, leading to partial opening of the structure.

## Telomerase translocation leads to complete unfolding of G-quadruplex structure

The second step of telomerase-mediated unfolding of G4 DNA requires telomerase catalytic activity (*Figure 5C–E*). To probe the mechanism for this, we incubated telomerase with F-22G3 in the presence of subsets of dNTPs. The first three nucleotides that are incorporated by telomerase at the 3' end of F-22G3 are dTTP, dATP and dGTP, as dictated by the telomerase RNA template sequence (see *Figure 6A,E and I*). We first carried out extension reactions in the presence of only ddTTP, a chain terminator that inhibits further elongation of the 3' end after its incorporation, and no other nucleotides (*Figure 6A*). Under these conditions, F-22G3 exhibited a FRET drop from ~0.5 to~0.3 and remained in the ~0.3 FRET state over the remainder of the observation time window (*Figure 6B–D* and *Figure 6—figure supplement 1*). A similar change in FRET from ~0.5 to~0.3 was observed when the only nucleotides in the reaction were dTTP and ddATP (*Figure 6F–H* and *Figure 6—figure supplement 1*). However, in the presence of dTTP, dATP and ddGTP, a second step-wise drop in FRET was exhibited by F-22G3, from ~0.3 to~0.15 (*Figure 6J–L* and *Figure 6—figure supplement 1*). These data demonstrate that complete G4 unfolding occurs after the addition of three nucleotides complementary to the template; at this point, the template boundary is reached and translocation of the DNA to the 3' region of the template is likely to occur (*Figure 6I*).

## Stable intermolecular, parallel G4 is unfolded and extended by telomerase using a similar mechanism

We next asked whether telomerase-mediated unfolding of intermolecular G4 occurs via a similar mechanism to that of intramolecular G4. To address this, we performed smFRET experiments in the presence of subsets of dNTPs and observed the change in FRET signal displayed by F-[7GGT]$_4$ over time (*Figure 6—figure supplement 2*). The FRET signal dropped in a single step, from ~0.6 to~0.4, in the presence of telomerase and either ddTTP alone or dTTP and ddATP. However, in the presence of telomerase and dTTP, dATP and ddGTP, a two-step decrease was observed, from a high FRET state (~0.6) to ~0.4 and then to the lowest FRET state (~0.2) (*Figure 6—figure supplements 2 and 3*). Overall, these data demonstrate that telomerase binds and unfolds parallel G4 using a similar mechanism, whether the strand topology is intramolecular or intermolecular.

## Ligand stabilization of intramolecular parallel G4 partially inhibits but does not prevent G4 unwinding by telomerase

Antiparallel or hybrid telomeric G4 are not efficiently used as substrates by telomerase, and their stabilization with small molecule G4-binding ligands can further decrease the ability of telomerase to extend them (*De Cian et al., 2007a*; *Zahler et al., 1991*; *Zaug et al., 2005*). We therefore sought to determine whether a ligand-mediated increase in stability of parallel G4 affects their extension by telomerase. To this end, we used three different G4-stabilizing compounds: the porphyrin N-methyl mesoporphyrin IX (NMM) (*Arthanari et al., 1998*; *Nicoludis et al., 2012a*), the berberine derivative SST16 (*Samosorn, 2016*; *Samosorn et al., 2009*) and the bisquinolinium compound PhenDC3 (*Chung et al., 2014*; *De Cian et al., 2007b*; *Figure 7—figure supplement 1*). CD spectroscopy confirmed that none of the ligands substantially changed the overall parallel G4 conformation of F-22G3

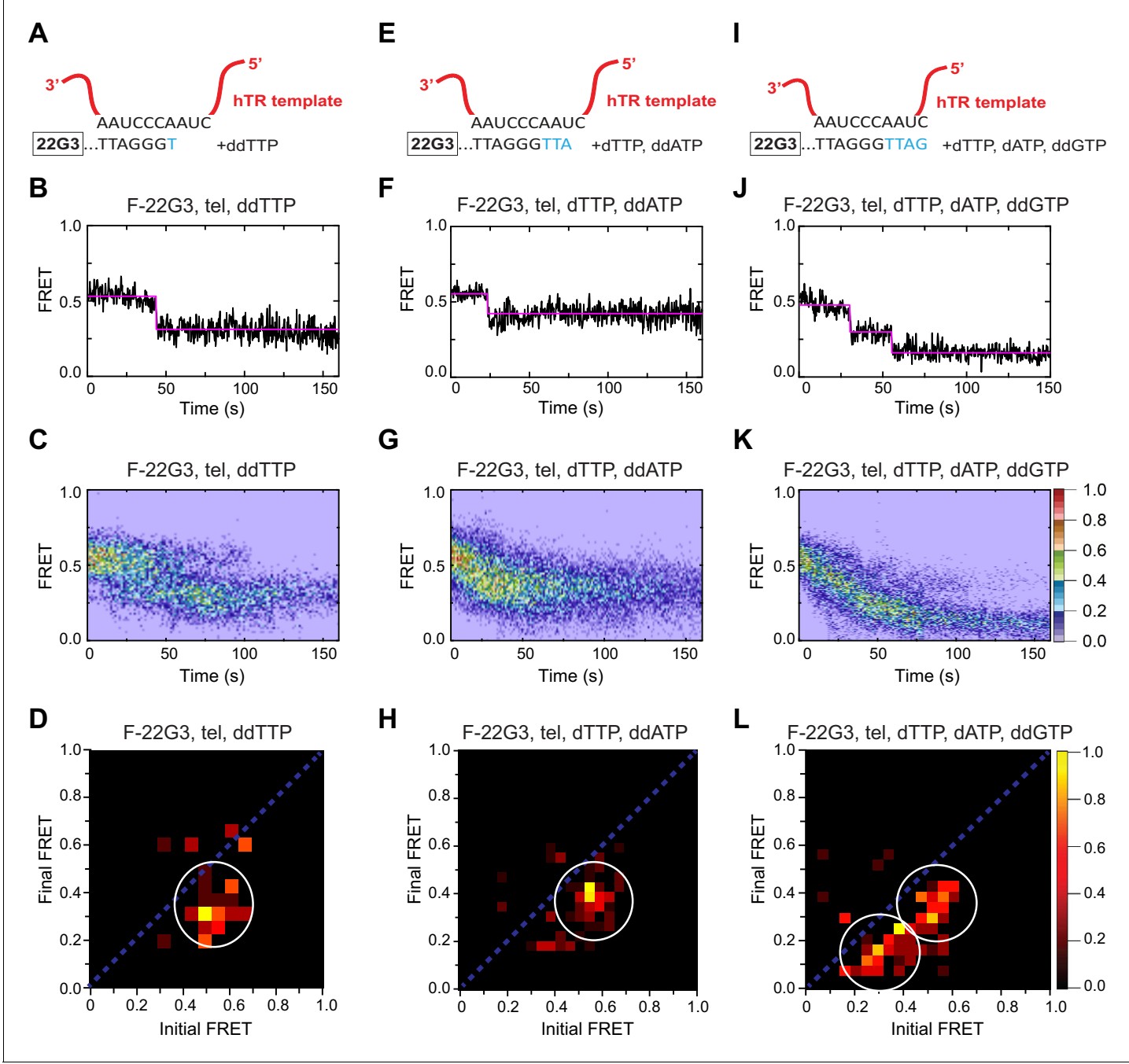

**Figure 6.** Telomerase translocation leads to complete G4 unfolding. (A), (E), (I) Schematic diagrams showing alignment of the telomerase template RNA with 22G3 DNA and template-directed incorporation of ddTTP (A), dTTP followed by ddATP (E) or dTTP, dATP and ddGTP (I). (B), (F), (J) Examples of individual F-22G3 FRET trajectories in the presence of telomerase and the indicated combinations of nucleotides. (C), (G), (K) Heat maps of the distribution of FRET intensities over 0–150 s in 80, 90 and 82 molecules, respectively, in the presence of telomerase and the indicated combinations of nucleotides. All plots include molecules collected in 4–6 independent experiments. For color key, see panel (K). (D), (H), (L) TDPs showing the changes between the initial and final FRET values of F-22G3 in the presence of telomerase and the indicated combinations of nucleotides (n = 80, 90 and 82 molecules, respectively). For color key, see panel (L).

The online version of this article includes the following source data and figure supplement(s) for figure 6:

**Figure supplement 1.** Quantitation of data in *Figure 6*.

**Figure supplement 1—source data 1.** Source data for the graph shown in *Figure 6—figure supplement 1*.

**Figure supplement 2.** Telomerase unfolds intra- and intermolecular G4 using same mechanism.

**Figure supplement 3.** Quantitation of data in *Figure 6—figure supplement 2*.

*Figure 6 continued on next page*

*Figure 6 continued*

**Figure supplement 3—source data 1.** Source data for the graph shown in *Figure 6—figure supplement 3*.

(*Figure 7—figure supplement 2A–C*), and melting assays showed dramatic thermal stabilization of this G4 upon binding by all three ligands ($\Delta T_m$ of >25°C (NMM), +11°C (SST16) and >25°C (PhenDC3), under the conditions detailed in *Figure 7—figure supplement 2D–F*). SST16 caused a decrease in signal of the peak at 260 nm that may be attributable to the association between the ligand and the G-quartets, resulting in slight changes in stacking interactions without a change in overall topology (*Ghosh et al., 2013*; *Gray et al., 2008*). The ligands did not change the steady smFRET signal of F-22G3 in the absence of telomerase (*Figure 7—figure supplement 3*). In the presence of telomerase, dNTPs and each of the three ligands, the number of molecules experiencing a two-step FRET decrease was reduced by about 2-fold, and the number of molecules with no FRET changes doubled (*Figure 7A–J*). This suggests that the ligands reduce binding of telomerase to this G4, preventing induction of the first step of FRET decrease. Nevertheless, 25–30% of molecules showed the same FRET decrease from ~0.5 to~0.3 and then to ~0.15 as in the absence of ligands (*Figure 7A–J*). The rates of telomerase-mediated unfolding of NMM-, SST16- and PhenDC3-stabilized F-22G3 were 0.071 ± 0.014 s$^{-1}$ ($N$ = 2.50 ± 0.04), 0.096 ± 0.01 s$^{-1}$ ($N$ = 2.6 ± 0.1), and 0.098 ± 0.015 s$^{-1}$ ($N$ = 2.60 ± 0.05), respectively (mean ± SEM; *Figure 7K*, *Figure 7—figure supplement 4*); in the case of NMM, this was significantly slower (p=0.0118; Student's t-test) than in the absence of ligands (0.12 ± 0.013 s$^{-1}$). At least in the case of NMM, the reduced rate of G4 unfolding provides evidence that most or all of these molecules were bound by ligands, but that ligand presence slowed the rate of their unfolding by telomerase. The very dramatic thermal stabilization of the G4 in the presence of PhenDC3 (*Figure 7—figure supplement 2F*) also suggests that a majority of molecules were bound by this compound. However, we cannot rule out that some of the molecules unwound by telomerase were not bound by the ligands.

Consistent with these data, all three ligands partially inhibited telomerase extension of F-22G3 when incubated at the same concentrations in ensemble telomerase activity assays (*Figure 7—figure supplement 5*). PhenDC3 resulted in a more dramatic decrease in activity than in G4 unfolding measured by smFRET, but it also caused a substantial loss of activity on a linear DNA substrate, consistent with its previously-described effects on telomerase in a manner independent of G4 (*De Cian et al., 2007a*).

Together, these data suggest that telomerase unfolding and extension of an intramolecular parallel G4 are partially inhibited by ligand stabilization; nevertheless, telomerase is able to overcome this stabilization and unwind a substantial proportion (25–30%) of molecules (*Figure 7J*).

## Ligand stabilization of intermolecular parallel G4 does not inhibit telomerase activity or telomerase-mediated G4 unfolding

To determine the generality of the ability of telomerase to extend G4 in the presence of stabilizing ligands, we also incubated [7GGT]$_4$ with the same three G4 ligands. CD spectroscopy confirmed that none of the ligands substantially changed the parallel G4 conformation of [7GGT]$_4$ (*Figure 8—figure supplement 1A–C*). Melting assays showed dramatic thermal stabilization of [7GGT]$_4$ upon binding by all three ligands ($\Delta T_m$ of +13°C (NMM), +21°C (SST16) and +26°C (PhenDC3), under the conditions detailed in *Figure 8—figure supplement 1D–F*). Telomerase activity assays were carried out at the same concentrations of [7GGT]$_4$ and each ligand as used in CD analyses; surprisingly, stabilization of the G-quadruplex did not inhibit its extension by telomerase (*Figure 8A–D*). As previously described (*De Cian et al., 2007a*), PhenDC3 caused a decrease in enzyme processivity after the addition of 4 telomeric repeats to either a linear or a G4 substrate (*Figure 8C*), most likely resulting from stabilization of the G-quadruplexes that have been demonstrated to occur within telomerase product DNA (*Jansson et al., 2019*; *Patrick et al., 2020*). Concentrations of PhenDC3 higher than 1 µM caused dramatic inhibition of activity (*Figure 8—figure supplement 2C*), but again this effect was observed with both linear and G4 substrates, indicating G4-independent direct inhibition of telomerase by PhenDC3, as previously described (*De Cian et al., 2007a*). Neither NMM nor SST16 inhibited extension of linear or G4 substrates at any concentration used (*Figure 8—figure*

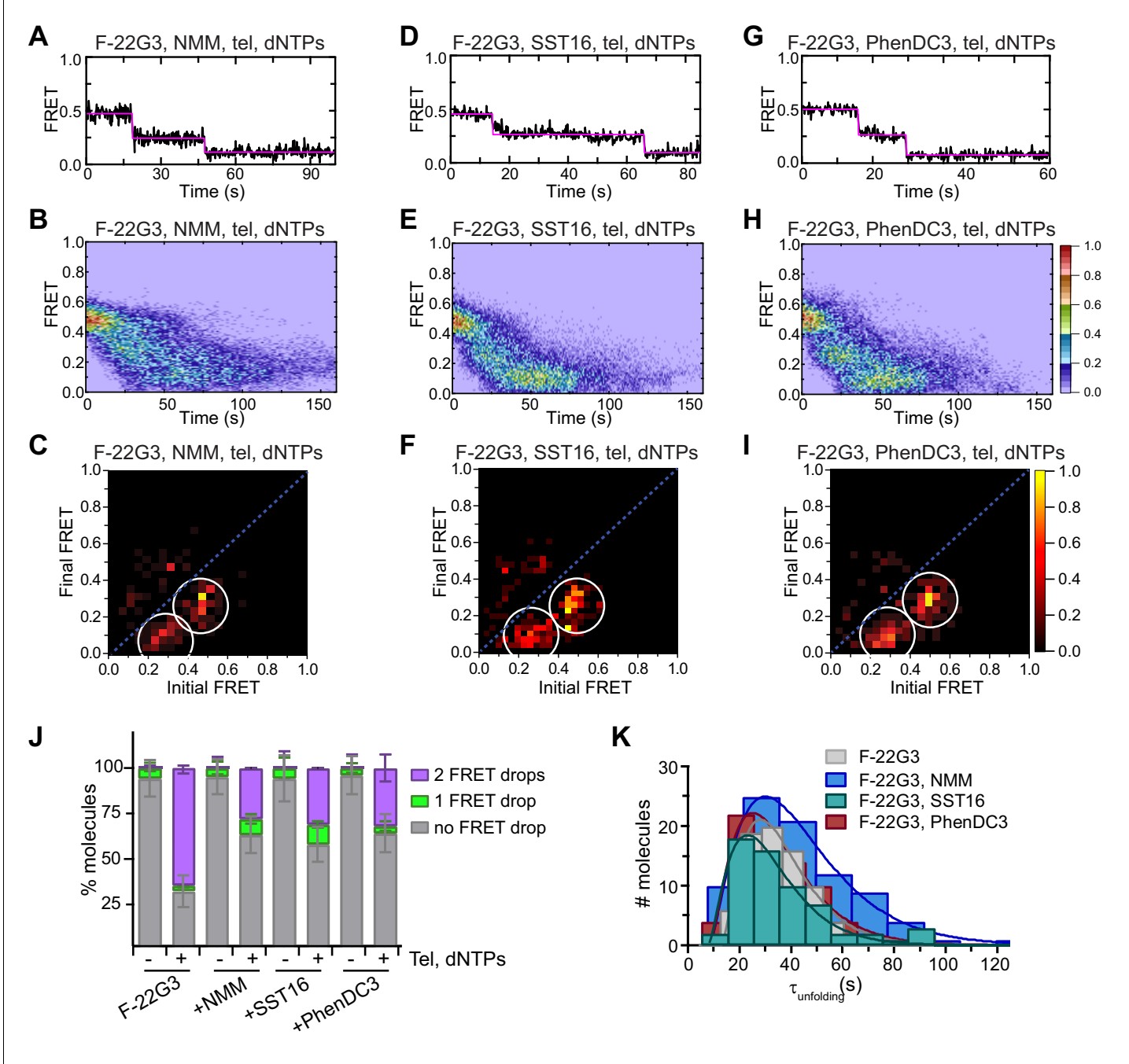

**Figure 7.** Partial inhibition of unfolding of F-22G3 by ligands NMM, SST16 and PhenDC3. (A – I) Representative FRET trajectories, heat maps and transition density plots of F-22G3 in the presence of NMM (A – C; 800 µM in folding reaction), SST16 (D – F; 5 µM) or PhenDC3 (G – I; 1 µM), in the presence of telomerase and dNTPs; n = 84 (B,C), 60 (E,F), and 77 (H,I) molecules, collected in 2–4 independent experiments. (J) Plot of the percentage of F-22G3 molecules showing no change in FRET, a single FRET drop, or a two-step FRET drop, when incubated with telomerase, dNTPs and the indicated ligands at the concentrations shown in (A – I); error bars represent sample size/√total population. (K) The unfolding rate of F-22G3 in the presence of the above concentrations of NMM, SST16, PhenDC3 or no ligand, calculated by fitting the dwell time distributions of the intermediate FRET state to a gamma distribution. Rates and curve fitting parameters are in *Figure 7—figure supplement 4*.

The online version of this article includes the following source data and figure supplement(s) for figure 7:

**Source data 1.** Source data for the graph shown in *Figure 7J*.
**Figure supplement 1.** Structures of G4-stabilizing ligands used in this study.
**Figure supplement 2.** Demonstration of ligand-mediated stabilization of F-22G3.
**Figure supplement 3.** Effects of ligands on F-22G3 FRET.
*Figure 7 continued on next page*

*Figure 7 continued*

**Figure supplement 4.** Curve fitting parameters of the curves in *Figure 7K*.
**Figure supplement 5.** Telomerase extension assays of F-22G3 in the presence of G4 ligands.

*supplement 2A and B*). Thus, substantial stabilization of a parallel intermolecular G4 by three different ligands did not inhibit the ability of telomerase to use the G4 as a substrate.

To further understand this effect at the molecular level we performed smFRET experiments to visualize F-[7GGT]$_4$ stabilized by NMM, SST16 or PhenDC3 in the presence of telomerase, with or without dNTPs. Ligand-stabilized F-[7GGT]$_4$ exhibited a constant FRET signal at 0.57 (*Figure 9—figure supplement 1*), but in the presence of telomerase, a single-step FRET decrease from 0.57 to 0.36 was observed in a majority of single-molecule traces (*Figure 9J*, *Figure 9—figure supplement 2*). These data suggest that ligand binding to F-[7GGT]$_4$ did not prevent telomerase from inducing a conformational change in the G4, as occurs in the absence of ligand. When F-[7GGT]$_4$ was incubated with both telomerase and dNTPs in the presence of each ligand, most molecules showed a unidirectional two-step decrease in FRET value, from 0.57 to 0.36, and then to 0.19 (*Figure 9A–J*). We assessed the F-[7GGT]$_4$ unfolding rate in the presence of the ligands by measuring the dwell time of each molecule in the transient intermediate state (0.36 FRET) and fitting time distributions with a gamma distribution (*Figure 9K*, *Figure 9—figure supplement 3*). The rates of telomerase-mediated unfolding of NMM-, SST16- and PhenDC3-stabilized G4 were $0.090 \pm 0.017$ s$^{-1}$ ($N = 2.40 \pm 0.04$), $0.28 \pm 0.07$ s$^{-1}$ ($N = 3.7 \pm 0.1$), and $0.15 \pm 0.05$ s$^{-1}$ ($N = 2.9 \pm 0.1$), respectively (mean $\pm$ SEM). Thus, while neither PhenDC3 nor NMM resulted in a significant change in the unfolding rate of this G-quadruplex, in the presence of SST16 unfolding occurred significantly faster ($p=0.021$; Student's t-test) than in the absence of ligands ($0.073 \pm 0.013$ s$^{-1}$). Together, these data demonstrate that neither unwinding nor extension of parallel intermolecular G4 by telomerase were inhibited by ligand-mediated stabilization of the G4.

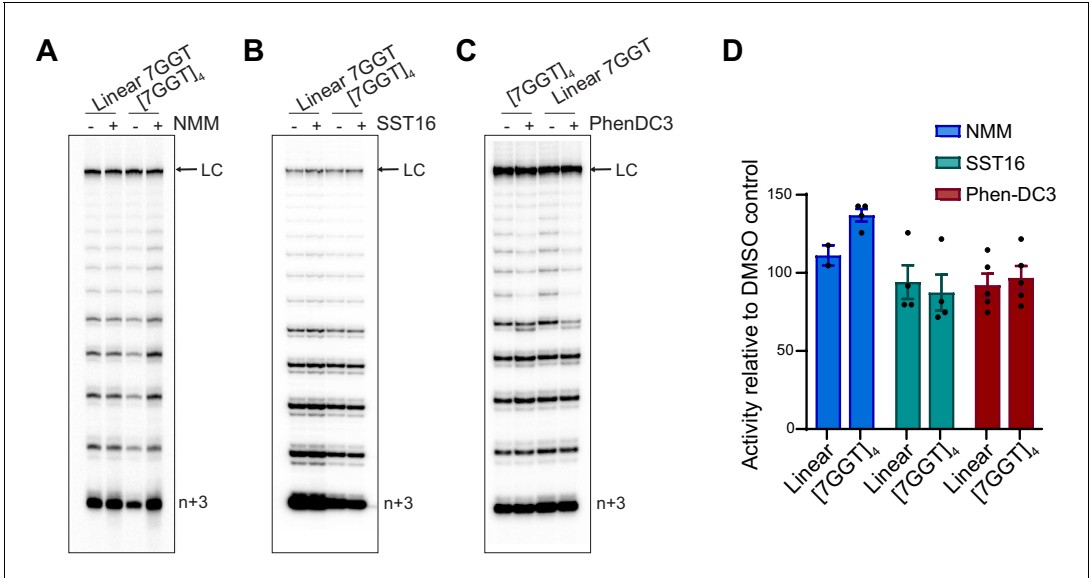

**Figure 8.** Telomerase extends ligand-stabilized tetrameric parallel G4. (**A**), (**B**), (**C**) Telomerase extension assays using 1 µM of [7GGT]$_4$ or a linear 7-mer control (or 250 nM in the reactions with PhenDC3), in the presence or absence of 40 µM NMM (**A**), 100 µM SST16 (**B**) or 1 µM PhenDC3 (**C**). For the reactions with NMM, linear 7GGT (1 mM) was incubated with 10 mM NMM prior to G4 folding, and the G4-ligand complex diluted 250-fold for the activity assay. n+3 indicates the position of the product with the first three nucleotides incorporated. LC: 5´-$^{32}$P-labeled synthetic 100 nt DNA used as a recovery/loading control. (**D**) Quantitation of activity in (**A–C**), normalized to the solvent (DMSO) control for each reaction. Graph shows mean $\pm$ SEM, n = 2–5 independent experiments.

The online version of this article includes the following figure supplement(s) for figure 8:

**Figure supplement 1.** Demonstration of ligand-mediated stabilization of F-22G3.
**Figure supplement 2.** Telomerase extension of [7GGT]$_4$ in the presence of titrations of G4 ligands.

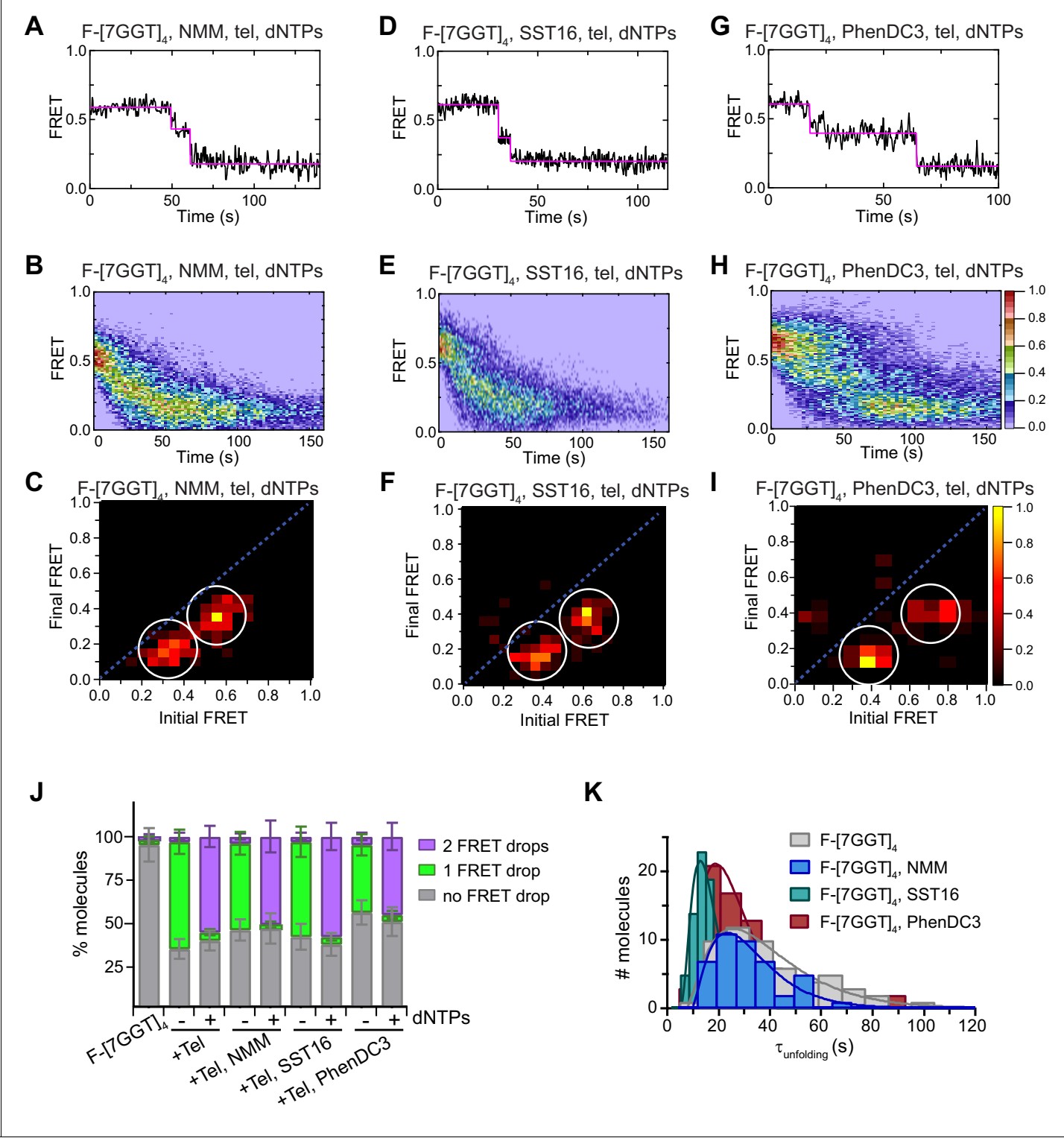

**Figure 9.** Telomerase unfolds ligand-stabilized tetrameric parallel G4. (A), (D), (G) Representative single-molecule FRET trajectories of NMM-stabilized (A; 10 mM in folding reaction), SST16-stabilized (D; 100 µM) or PhenDC3-stabilized (G; 1 µM) F-[7GGT]$_4$ in the presence of telomerase and dNTPs. (B), (E), (H) Heat maps of the distribution of FRET intensities over 0–150 s in the presence of telomerase, dNTPs and either NMM (n = 63) (B), SST16 (n = 90) (E) or PhenDC3 (n = 76) (H). All plots include molecules collected in 4–6 independent experiments. For color key, see panel (H). (C), (F), (I) TDPs showing the changes in FRET value of 70, 80 and 58 molecules from the experiments in (A) to (I). (J) Plot of the percentage of F-[7GGT]$_4$ molecules showing no change in FRET, a single FRET drop, or a two-step FRET drop, when incubated with telomerase and the indicated ligands at the

*Figure 9 continued on next page*

*Figure 9 continued*

concentrations shown in (A) – (I); error bars represent sample size/$\sqrt{\text{total population}}$. (K) The unfolding rate of F-[7GGT]$_4$ in the presence of the above concentrations of NMM, SST16, PhenDC3 or no ligand, calculated by fitting the dwell time distributions of the intermediate FRET state to a gamma distribution. Rates and curve fitting parameters are in *Figure 9—figure supplement 3*.

The online version of this article includes the following source data and figure supplement(s) for figure 9:

**Source data 1.** Source data for the graph shown in *Figure 9J*.
**Figure supplement 1.** Effects of ligands on F-[7GGT]$_4$ FRET.
**Figure supplement 2.** Effect of ligands on FRET of F-[7GGT]$_4$ in the presence of telomerase.
**Figure supplement 3.** Curve fitting parameters of the curves in *Figure 9K*.

## Telomerase can potentially accommodate a G4 substrate after protein conformational changes

Our data demonstrate that telomerase extends G4 substrates while they are partially structured (data herein, and *Moye et al., 2015*); however, they are bulkier than their single-stranded DNA counterparts, so how the active site of telomerase accommodates G4 has been an open question. To address this, we performed molecular modeling of binding of the tetrameric G4 [7GGT]$_4$ to human telomerase.

We had previously generated a homology model of the hTERT reverse transcriptase domain using the published crystal structure of *Tribolium castaneum* TERT (*Gillis et al., 2008*; *Tomlinson et al., 2015*). More recently, the structure of the *Tetrahymena thermophila* enzyme has been solved by cryo-electron microscopy (cryo-EM) (*Jiang et al., 2018*), and there is also a medium-resolution cryo-EM structure of human telomerase available (*Nguyen et al., 2018*). We therefore used these to generate a model of the whole hTERT protein (see Methods). We then took the NMR structure of [7GGT]$_4$ (*Gavathiotis and Searle, 2003*), and introduced torsion between the T and A positions, to model one strand flipping out in order to hybridize with a 10 nt RNA molecule with the sequence of the human telomerase template. The position of a DNA-RNA hybrid in the *Tetrahymena* cryo-EM structure was used to position this G4-RNA into the telomerase active site, with the 3' end of the flipped strand at the trio of catalytic aspartates (*Figure 10A*). As expected, the G4-RNA clashed with part of the hTERT protein, the C-terminal extension (CTE). However, others have reported that the CTE may pivot around a region called the 'beta linker' (*Wu et al., 2017*; *Yang and Lee, 2015*); thus, it seemed reasonable that this linker may lengthen to accommodate the quadruplex. This is also consistent with our finding that the affinity of telomerase for nucleotides is decreased in the presence of a G4 substrate relative to a linear substrate (*Moye et al., 2015*; *Oganesian et al., 2007*), suggesting that the conformation of the active site is perturbed when bound to the bulkier G4. The torsions of the beta linker amino acids (Arg938 and Thr939) were therefore altered to enable an extended conformation, which was compatible with binding of the G4-RNA in the active site (*Figure 10A*).

Molecular dynamic simulations were performed for 10 ns to obtain a minimum energy conformation. Interestingly, if allowed to continue for 100 ns, the simulations showed the 5' ends of the G4 DNA strands moving apart from one another (*Figure 10B*, strands in blue). While this represents an in silico simulation rather than direct structural observation, it suggests that binding of this G4 to telomerase may result in an increase in distance between the dyes, consistent with the observed partial reduction in FRET.

## Discussion

Previously, we have shown that highly purified human telomerase can disrupt parallel intermolecular G4 structures and then extend them in a processive manner (*Moye et al., 2015*). Here, we provide the first mechanistic details of telomerase resolution of parallel G4 structures. Our data demonstrate that G4 unfolding occurs in three major steps: i) the telomerase template RNA invades the G-quartets in order to hybridize with the G-rich DNA, causing partial opening of the G4 structure; ii) telomerase extends the 3' end of this partially unwound structure, adding single nucleotides according to the RNA template, and iii) translocation of telomerase once nucleotide addition has reached the template boundary is likely to trigger complete unfolding of the G4 structure (*Figure 10C*). It has

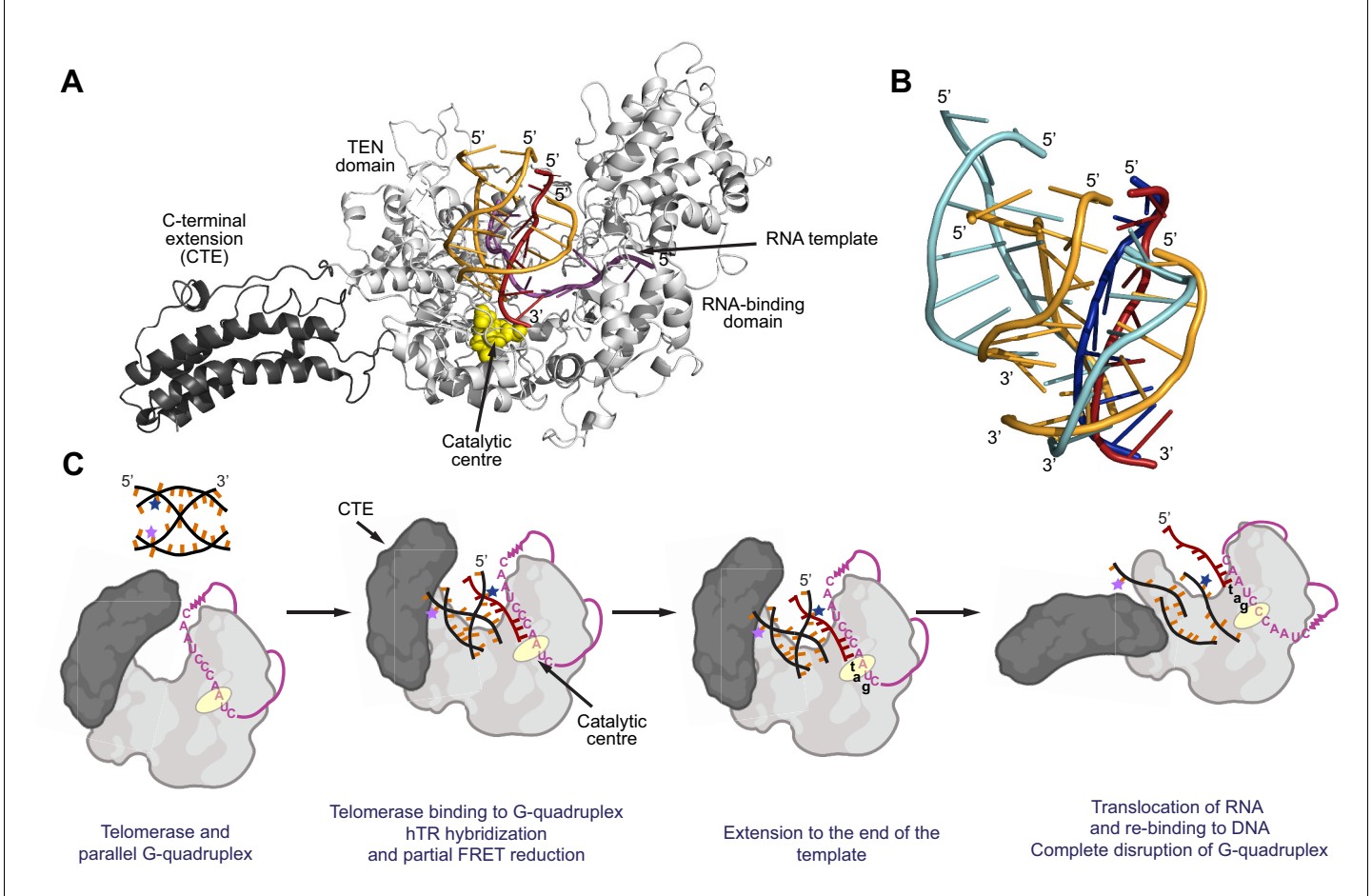

**Figure 10.** Model for binding and extension of parallel G-quadruplexes by telomerase. (**A**) Model of an extended conformation of the human telomerase reverse transcriptase (hTERT; grey) with a 10 nt RNA template (magenta) and the [7GGT]₄ G-quadruplex (orange and dark red) docked into the active site. The extended CTE is highlighted in dark grey; the catalytic aspartate residues (Asp 712, 868, 869) are shown as yellow spheres. A single strand of the G4 (dark red) is flipped out of the structure in order for it to interact with the RNA at the catalytic center of the reverse transcriptase. Note that the conformation shown has a fully-extended CTE, but intermediate angles of pivoting are also compatible with binding of [7GGT]₄. (**B**) Positions of the strands of [7GGT]₄ after 10 ns (orange, dark red strands) or 100 ns (blue strands) of energy minimization with molecular dynamics. The dark red/dark blue strand is the one hybridizing with RNA. The modeling was performed with the G4 in the active site of telomerase; the protein and RNA are not shown. (**C**) A proposed model for parallel G4 extension and unfolding by human telomerase. See text for details.

previously been difficult to distinguish whether it is the parallel or intermolecular nature of G4 structures that allows their recognition by telomerase. Here, we overcome this difficulty by exploiting the stabilization of intramolecular parallel G4 conferred by the modified nucleotide 2′F-araG (*Abou Assi et al., 2017*), enabling the demonstration that inter- and intramolecular parallel G4 are extended by telomerase using the same 3-step mechanism. These data reveal human telomerase to be a parallel G4 resolvase.

Several other proteins have been shown to assist telomerase extension of telomeric DNA by resolving antiparallel G4 in the substrate, including telomeric protein POT1 (*Hwang et al., 2014*; *Zaug et al., 2005*), a splice variant of hnRNP A2 (*Wang et al., 2012*), and the ciliate protein StyRecQL (*Paeschke et al., 2008*; *Postberg et al., 2012*). In contrast, we have verified that highly purified telomerase, free of contaminating helicases or other proteins, is able to extend [7GGT]₄ (*Moye et al., 2015*). Therefore, we conclude that telomerase itself has the ability to unwind parallel G4, without the assistance of interacting helicases.

Telomerase partially unfolds parallel G4, independently of telomerase catalytic activity, as indicated by the drop in FRET signal upon telomerase association with G4 (*Figure 5A–E*). The resulting stable intermediate FRET value is consistent with a partial separation of the FRET dyes in the

absence of complete strand dissociation. We interpret these data to indicate that binding of telomerase RNA to the nucleotides at the 3' end of one DNA strand allows RNA-DNA hybridization to occur while resulting in a conformational perturbation of the 5' end of the G4, but the dyes are still sufficiently close together to give a non-zero FRET signal; molecular modeling supported this interpretation (*Figure 10B and C*). To investigate whether the template RNA of telomerase plays a critical role in this initial G4 unfolding, we carried out experiments in the presence of a short RNA oligonucleotide with a sequence complementary to the telomere repeat, mimicking the template region of hTR. This RNA molecule was able to partially unfold the structure in a similar manner to the whole enzyme, albeit with lower affinity. This observation suggests that it is the RNA component of telomerase, rather than the hTERT protein, that promotes the initial unwinding of parallel G4, followed by hybridization to the 3' end. A DNA oligonucleotide of identical sequence did not promote any G4 unwinding (*Figure 5H–J*); if oligonucleotide binding was due to passive trapping of a spontaneously-unfolding G4, this would also have been observed in the presence of a DNA C-strand. Previous studies with relatively unstable antiparallel telomeric G4s in Na$^+$ have observed passive trapping of the unfolded G4 by a peptide nucleic acid (PNA) oligonucleotide; the unfolding was zero order with respect to the PNA molecule, suggesting that the initial step is a rate-determining internal rearrangement of the quadruplex (*Green et al., 2003*). In contrast, both parallel G4s used in our experiments were highly stable; neither demonstrated any spontaneous unwinding during the observation time of the smFRET experiments, and in our previous work we have shown that the parallel tetrameric G4 did not show any unwinding over the time scale of hours (*Moye et al., 2015*).Telomeric protein POT1 unwinds a G4 through a conformational selection mechanism in which G4 unwinding occurs prior to POT1 binding (*Chaires et al., 2020*); each of the OB folds of two POT1 molecules then binds to one telomeric repeat of the 4-repeat G4 in a stepwise manner (*Hwang et al., 2012*). For telomerase, we favour a model in which the enzyme binds the G4 while it is still structured, followed by invasion by the RNA template; confirmation of this model will require further kinetic analyses. A similar template-mimicking RNA was unable to form a DNA-RNA hybrid with an antiparallel G4 of similar stability to the parallel G4 used in the present study (*Birrento et al., 2015*), demonstrating that this is a specific interaction between this RNA and parallel G4 structures. Single-molecule FRET studies have illuminated the mechanisms used by other G4-unwinding proteins; for example, the helicase RHAU/DHX36 was shown to stack on the top of the G4 plane and bind to the exposed 3' single stranded DNA tail (*Chen et al., 2018*), whereas the replication protein RPA has been proposed to bind to G4 structures via their exposed loops (*Ray et al., 2013*). In contrast, our data demonstrate that telomerase possesses a unique mechanism of G4 resolution, mediated by its integral RNA subunit.

Telomerase extends the 3' end of parallel G4 while the G4 remains partially structured, as demonstrated by the intermediate FRET state remaining unchanged until the hTR template boundary was reached (*Figure 6*). Although these experiments involved incorporation of a chain terminating ddGTP, it has been demonstrated by smFRET that ddGTP incorporation is compatible with subsequent DNA realignment and rehybridization with the upstream region of the template (*Parks and Stone, 2014*); it is therefore likely that conformational changes involved in the translocation process result in complete G4 unfolding, represented by a second step-wise drop in FRET signal. One possible explanation for this observation is that the G4 structure needs to pass through the telomerase DNA-binding cleft during the conformation changes required for telomerase translocation, and the resulting steric hindrance results in complete dissociation of the parallel telomeric G4. Our molecular modeling supports an alternative scenario, in which an hTERT protein conformational change during translocation results in opening of the active site, allowing the strands of the G4 to move away from each other (*Figure 10C*). This would be consistent with a 'pivoting' of the CTE away from the catalytic center that has been proposed to allow room for DNA-RNA dissociation and movement during translocation (*Wu et al., 2017*; *Yang and Lee, 2015*). In the case of a G4 substrate, it is possible that slight pivoting of the CTE is needed to allow the structure to bind in the enzyme active site (*Figure 10C*, middle panels). After nucleotide addition, when the end of the template is reached, this may act as a trigger for full pivoting of the CTE to allow translocation to occur (*Wu et al., 2017*; *Figure 10C*, right panel; the model in *Figure 10A* also shows the fully-extended structure). This may allow full dissociation of the strands of the G4, resulting in a drop in FRET to its lowest level.

Quantitation of the rates of the transition from the first FRET drop to the second support the notion that telomerase-mediated G4 unfolding is a complex multistep process, consisting of at least

five kinetic steps. This is consistent with the requirement for extension of the DNA to the template boundary and translocation, processes which involve many kinetic substeps: nucleotide binding (*Tomlinson et al., 2015*), polymerization at each template position (*Tomlinson et al., 2015*), DNA primer realignment with the upstream region of the RNA (*Parks and Stone, 2014*), and conformational rearrangements to reposition the DNA-RNA hybrid into the active site (*Parks and Stone, 2014*).

We used the G4-stabilizing ligands NMM, SST16 and PhenDC3, all of which bind to parallel G4 (*Lacroix et al., 2011*; *Nicoludis et al., 2012a*; *Samosorn et al., 2009*), to understand whether telomerase extension of these structures can be modulated. Despite substantial thermal stabilization of the G4, these ligands did not inhibit telomerase extension of intermolecular parallel telomeric G4, with one of them instead slightly accelerating G4 unfolding. The ligands did reduce the number of intramolecular parallel G4 molecules unwound by telomerase; nevertheless, some molecules were unwound by telomerase despite ligand binding, using the same two-step mechanism as in the absence of ligands. The fact that telomerase was able to extend parallel G4 even after substantial thermal stabilization of the structure using either small molecule ligands (*Figures 7–9*) or chemical modification (*Figure 1*) provides evidence that thermal stabilization does not provide a barrier to telomerase-mediated unwinding.

NMM has been demonstrated to bind to parallel telomeric G4 by stacking on the outermost G-quartets (*Nicoludis et al., 2012b*), which would be predicted to interfere with telomerase extension. This observation suggests that, like other proteins such as RHAU helicase (*Gueddouda et al., 2017*; *Tippana et al., 2016*), telomerase is able to dislodge ligands from the tetramolecular G4 structure while unfolding it. This displacement activity is structure dependent, since telomerase is less able to dislodge ligands from a parallel intramolecular G4, possibly due to the presence of DNA loops. Such a displacement mechanism would have implications for interpreting the biological effects of G4-stabilizing ligands in cell-based studies, since any effects on telomerase action and telomere length will depend on the conformational specificity of the ligand.

There is growing evidence for G4 formation at telomeres (*Biffi et al., 2013*; *Moye et al., 2015*; *Murat and Balasubramanian, 2014*; *Paeschke et al., 2005*; *Schaffitzel et al., 2001*; *Zhang et al., 2010*). In human cells, it is not known if they form at the 3' end of the telomeric overhang, internally, or a combination of both, or which conformation of G4 occurs at these locations in vivo. While in-cell NMR studies of transfected telomeric oligonucleotides detected hybrid and antiparallel structures (*Bao et al., 2019*), a study using parallel-specific antibodies reported detection of parallel telomeric G-quadruplexes in human cells (*Liu et al., 2016*) and a parallel-specific ligand induces G4 formation at human telomeres (*Moye et al., 2015*). It is also possible that different conformations exist at different telomeric overhangs within the same cell. Intermolecular parallel G4 may also form between telomeric overhangs; it has been suggested that tetrameric parallel G-quadruplexes could be responsible for the correct alignment of four chromatids during meiosis (*Sen and Gilbert, 1988*). Regardless of this uncertainty, our data unequivocally demonstrate that human telomerase interacts directly with parallel G4 structures and resolves them. The conserved ability of human and ciliate telomerase enzymes to unwind these structures suggests that parallel G-quadruplexes form at telomeric overhangs in vivo, and do not form a barrier to telomerase extension.

## Materials and methods

**Key resources table**

| Reagent type (species) or resource | Designation | Source or reference | Identifiers | Additional information |
|---|---|---|---|---|
| Cell line (*Homo sapiens*) | HEK293T (embryonic kidney, immortalized with adenovirus) | American Type Tissue Collection | Cat#: ATCC CRL-3216 | |
| Antibody | Anti-hTERT (sheep polyclonal) | Abbexa Ltd. | Cat#: abx120550 | IP (40 µg/ml) |
| Recombinant DNA reagent | pApex-CMV-hTERT (plasmid) | PMID:26158869 | | Dr Tracy Bryan, Dr Scott Cohen |

*Continued on next page*

*Continued*

| Reagent type (species) or resource | Designation | Source or reference | Identifiers | Additional information |
|---|---|---|---|---|
| Recombinant DNA reagent | pApex-CMV-dyskerin-U3-hTR (plasmid) | PMID:26158869 | | Dr Tracy Bryan, Dr Scott Cohen |
| Peptide, recombinant protein | hTERT amino acids 276–294 (peptide) | Abbexa Ltd. | Cat#: abx069990 | Sequence: ARPAEEATSL EGALSGTRH |
| Commercial assay or kit | Dynabeads M-280 Streptavidin | Thermo-Fisher | Cat#: 11206D | |
| Commercial assay or kit | NeutrAvidin Protein | Thermo-Fisher | Cat#: 31050 | |
| Commercial assay or kit | Unylinker CPG solid support | ChemGenes | Cat#: N-4000–05 | |
| Commercial assay or kit | Gel-Pak 2.5 Desalting Column | Glen Research | Cat#: 61-5025-25 | |
| Chemical compound, drug | AlexaFluor 555 NHS Ester | Thermo-Fisher | Cat#: A20009 | |
| Chemical compound, drug | (3-Aminopropyl) triethoxy silane | Alfa Aesar | Cat#: A10668 | |
| Chemical compound, drug | Biotin-PEG-SVA and mPEG-SVA | Laysan Bio | Cat#: BIO-PEG-SVA-5K-100MG and MPEG-SVA-5K-1g | |
| Chemical compound, drug | N-Methyl Mesoporphyrin IX (NMM) | Frontier Scientific | Cat#: NMM580 | |
| Chemical compound, drug | PhenDC3 | PMID:17260991 | | Dr Marie-Paule Teulade-Fichou |
| Chemical compound, drug | SST16 | PMID:19419877 | | Dr Siritron Samosorn |

## Oligonucleotides and G-quadruplex preparation

Most DNA oligonucleotides (*Supplementary file 1*) were purchased from Integrated DNA Technologies and RNA oligonucleotides from Dharmacon, with purification by high performance liquid chromatography (HPLC). Oligonucleotides 22G0, 22G3, and 22G0+tail were synthesized as previously described (*Abou Assi et al., 2017*). Synthesis of (C5-alkylamino)-dT-41G3 (the equivalent of 22G3 +tail prior to AlexaFluor 555 conjugation) was performed on an ABI 3400 DNA synthesizer (Applied Biosystems) at 1 µmol scale on Unylinker CPG solid support. Conjugation of (C5-alkylamino)-dT-41G3 to AlexaFluor 555 NHS Ester to produce the desired 22G3+tail oligonucleotide was achieved following the standard protocol by Sigma-Aldrich (Protocol for Conjugating NHS-Ester Modifications to Amino-Labeled Oligonucleotides). Briefly, a solution of (C5-alkylamino)-dT-41G3 (200 µL, 0.3 mM) in sodium tetraborate decahydrate buffer (0.091 M, pH 8.5) was combined with a solution of Alexa-Fluor 555 in anhydrous DMSO (50 µL, 8 mM) and the reaction mixture was left shaking at room temperature for 2.5 hr. Samples were evaporated to dryness and the 22G3+tail product was purified by anion exchange HPLC as described (*Abou Assi et al., 2017*). The peak of (C5-alkylamino)-dT-41G3 eluted between 24 and 26 min, while the desired product peak of 22G3+tail eluted between 27 and 28.5 min. Based on the area of the two peaks, the yield of the conjugation reaction is approximately 25%. The collected sample of 22G3+tail was desalted on Gel-Pak 2.5 Desalting Columns according to the manufacturer's protocol. The maximum absorbance of AlexaFluor 555 (555 nm) in the purified 22G3+tail was verified with UV-Vis spectroscopy. The mass of 22G3+tail was verified by high resolution liquid chromatography–mass spectrometry (LC–MS; 14,000.23 m/z).

Intramolecular G4 formation using 22G0 and derivatives (*Supplementary file 1*) was performed at a DNA concentration of 10 µM in 20 mM potassium phosphate, 70 mM potassium chloride pH 7, 1 mM MgCl$_2$, by heat denaturing for 10 min at 90℃, allowing the DNA to cool slowly (~1 hr) to 25℃ and equilibration at this temperature for 12–16 hr. Formation of fluorescently-labeled F-22G3 with a duplex tail (*Figure 1C*) involved incubation of 10 µM each of oligonucleotides 22G3+tail and 647-Strand2 (*Supplementary file 1*) under the same folding conditions as given for 22G0.

7GGT and its labeled derivatives were combined at a final concentration of 1 mM in K$^+$ hTel buffer (50 mM Tris-HCl, pH 8, 1 mM MgCl$_2$, 150 mM KCl) and heat denatured for 5 min at 95℃. They were allowed to cool slowly (~1 hr) to 25℃ and left to equilibrate at this temperature for 72 hr. Intermolecular G4 formation was confirmed by native gel electrophoresis followed by staining with SYBR Gold (Life Technologies), as described (*Moye et al., 2015*). DNA concentrations were determined by UV absorbance at 260 nm (extinction coefficients in *Supplementary file 1*). Concentrations of G-quadruplexes are given as the concentration of assembled complexes (i.e. taking strand stoichiometry into account). Folded G-quadruplexes were stored at 4℃ until use.

## Circular dichroism

Circular dichroism (CD) spectra were recorded at 25℃ on either an Aviv 215S or a JASCO J-810 CD spectrometer equipped with Peltier temperature controllers. G-quadruplex samples of the desired conformation were prepared at 250 nM - 20 µM in their folding buffers. Three to four scans were accumulated over the indicated wavelength ranges in a 0.1 cm or 1 cm path length cell. Parameters used with the Aviv CD spectrometer included a time constant of 100 ms, averaging time 1 s, sampling every 1 nm, and bandwidth 1 nm, while the JASCO CD spectrometer was used with a scan rate of 100 nm/min and a response time of 2.0 s. Buffers alone were also scanned and these spectra subtracted from the average scans for each sample. CD spectra were collected in units of millidegrees, normalized to the total species concentrations and expressed as molar ellipticity units (deg × cm$^2$ dmol$^{-1}$). Data were smoothed using the Savitzky-Golay function within the JASCO graphing software, or the smoothing function within GraphPad Prism. For thermal stability analysis, the samples were scanned using the above parameters, but with a fixed wavelength (260 nm) over increasing temperature (25℃ to 100℃), at a rate of ~1 ℃/min. For reactions including SST16 or PhenDC3 (prepared as described; *De Cian et al., 2007b*; *Samosorn et al., 2009*), the ligand was incubated with the folded DNA substrate at 25℃ for 30 min prior to CD spectroscopy. For reactions including NMM (Frontier Scientific, USA), the ligand was incubated with the DNA prior to G-quadruplex folding and the G4-ligand complex diluted prior to CD spectroscopy. Concentrations of ligands and G4 DNA used for each experiment are given in the figure legends.

## Preparation of telomerase

Human telomerase was overexpressed in HEK293T cells (identity verified by STR profiling, certified mycoplasma-free) and purified as described (*Moye et al., 2015*; *Tomlinson et al., 2017*). Briefly, plasmids encoding hTERT, hTR and dyskerin (available from the authors with an accompanying Materials Transfer Agreement) were transiently transfected into HEK293T cells growing in 20 L bioreactors using polyethylenimine and cells harvested 4 days later. Cell lysates were clarified, ribonucleoprotein complexes enriched with MgCl$_2$, and telomerase immunoprecipitated with a sheep polyclonal hTERT antibody, raised against hTERT amino acids 276–294 (ARPAEEATSLEGALSGTRH) (*Cohen and Reddel, 2008*). Telomerase was eluted by competitive elution with the same peptide in 20 mM HEPES-KOH (pH 8), 300 mM KCl, 2 mM MgCl$_2$, 0.1% v/v Triton X-100% and 10% v/v glycerol. Fractions were assayed for telomerase concentration by dot-blot northern against hTR (*Tomlinson et al., 2017*), and equal amounts of enzyme (~1.5 nM) used in each activity assay.

### Telomerase activity assays

The following reaction was prepared to give 20 µL per sample: 250 nM - 2 µM of the specified oligonucleotide (concentrations given in figure legends), 20 mM HEPES-KOH (pH 8), 2 mM MgCl$_2$, 150 mM KCl, 5 mM dithiothreitol, 1 mM spermidine-HCl, 0.1% v/v Triton X-100, 0.5 mM dTTP, 0.5 mM dATP, 4.6 µM nonradioactive dGTP and 0.33 µM [α-$^{32}$P]dGTP at 20 mCi mL$^{-1}$, 6000 Ci mmol$^{-1}$ (PerkinElmer Life Sciences). For reactions including SST16 or PhenDC3, the ligand was incubated with the folded DNA substrate at 25℃ for 30 min prior to adding other components. For reactions

including NMM, the ligand was incubated with the DNA prior to G-quadruplex folding; concentrations are given in figure legends. Telomerase activity assays were initiated by adding purified human telomerase to ~1.5 nM, and incubating at 37°C for 1 hr. The reaction was quenched by the addition of 20 mM EDTA and $1–2 \times 10^3$ cpm of a 5'-$^{32}$P-labeled synthetic 100-mer, 30-mer or 12-mer DNA (as indicated in figure legends) as an internal recovery standard. Products of telomerase extension were recovered as described, either with phenol/chloroform extraction followed by ethanol precipitation (*Moye et al., 2015*), or, for biotinylated substrates, by recovery with magnetic streptavidin beads (*Tomlinson et al., 2017*). The solution was heated at 90°C for 5 min, and 3 µL was electrophoresed over a 10% polyacrylamide sequencing gel (0.2 mm thick x 40 cm length x 35 cm width, 32-well comb) run in 1 × TBE/8 M urea at 85 W. The gel was transferred to filter paper, dried for 30 min at 80°C, exposed to a PhosphorImager screen, visualized on a Typhoon FLA9500 scanner (GE Healthcare Lifesciences) and analyzed using ImageQuant software.

## Single-molecule fluorescence imaging and data analysis

### Microscope setup for FRET imaging

A home-built objective-type total internal reflection fluorescence (TIRF) microscope based on an Olympus IX-71 model was used to record single-molecule movies. A Coherent Sapphire green (532 nm) laser was used to excite donor molecules at an angle of TIRF by focusing on a 100X oil immersed objective. FRET was measured by excitation with a 532 nm laser and the emissions at 565 and 665 nm were collected using a band pass filter at 560–600 nm and a long pass filter at 650 nm. Scattered light was removed by using a 560 nm long pass filter. AlexaFluor 555 and AlexaFluor 647 signals were separated by 638 nm dichroic using photometrics dual view (DV-2) and both signals were focused onto a charge-coupled device (CCD) camera (Hamamatsu C9 100–13), simultaneously. Data were collected at five frames per second.

### Sample preparation for FRET experiments

Quartz coverslips were treated with 100% ethanol and 1 mM KOH. Then, aminosilanization of coverslips was carried out in a 1% v/v (3-Aminopropyl)triethoxy silane solution in acetone. PEGylation was carried out by incubating a mixture of biotin-PEG-SVA and mPEG-SVA at a ratio of 1:20 prepared in 0.1 M NaHCO$_3$ solution on the top of a silanized coverslip for at least 3–4 hr. Finally, PEGylated coverslips were stored under dry nitrogen gas at −20°C.

### Single-molecule experiments

NeutrAvidin Protein solution was prepared in K$^+$ buffer (10 mM Tris-HCl (pH 8), 1 mM MgCl$_2$ and 150 mM KCl) and spread on the top of a dry PEGylated coverslip followed by a 10 min incubation. Sample flow chambers were created by sandwiching polydimentylsiloxane (PDMS) on top of the neutravidin coated coverslip. Then, blocking buffer (K$^+$ buffer with 1% Tween-20) was injected into the channel in order to reduce non-specific binding of proteins on the surface, followed by 10–15 min incubation. A 50 pM solution of biotinylated FRET G-quadruplex substrate was prepared in K$^+$ buffer and 200 µL injected into the flow chamber using a syringe pump (ProSense B.V.) followed by incubation for 10 min. Unbound sample was washed off in K$^+$ buffer. Movies were recorded at room temperature (20 ± 1°C) for 3–5 min in oxygen-scavenging system (OSS) consisting of protocatechuic acid (PCA, 2.5 mM) and protocatechuate-3,4-dioxigenase (PCD, 50 nM) to reduce photo-bleaching of the fluorophores, and 2 mM Trolox to reduce photo-blinking of dyes. For the Li$^+$ buffer substitution reactions, either inter- or intramolecular G4 substrates were immobilized on the coverslip surface in K$^+$ buffer followed by washing off excess of unbound molecules using K$^+$ buffer containing OSS. Then, Li$^+$ buffer (10 mM Tris-HCl (pH 8), 1 mM MgCl$_2$ and 150 mM LiCl) containing OSS was introduced into the flow cell to replace the K$^+$-buffer while monitoring the change in FRET experienced by G4 molecules in real time; movies were acquired until all acceptor molecules disappeared from the field of view. For experiments in the presence of enzyme, 200 µL of telomerase (0.5 nM; expressed and purified as described above) and/or dNTPs or ddNTPs (0.5 mM) were injected into the microscopic channel containing immobilized G4 DNA while the movie was recorded continuously. The buffer reached the microscopic channel at ~10 s and the movie was collected until all acceptor molecules photobleached. For reactions including SST16 or PhenDC3, the ligands were incubated with the folded DNA substrate at 25°C for 30 min prior to dilution for sample injection.

For reactions including NMM, the ligand was incubated with the DNA prior to G-quadruplex folding, and the G4-ligand complex diluted to 50 pM G4 prior to injection. Concentrations of DNA and ligands combined for each experiment are given in the figure legends.

## Data analysis

Single-molecule intensity time trajectories were generated in IDL and analyzed in MATLAB using custom-written scripts (software available at https://cplc.illinois.edu/research/tools). Approximate FRET value is measured as the ratio of acceptor intensity to the sum of the donor and acceptor intensities after correcting cross talk between donor and acceptor channels for both inter- and intramolecular G4s. Briefly, FRET pairs were identified through aligning the red (acceptor) and green (donor) channels by using highly fluorescent beads and subsequent co-localization of the donor/acceptor fluorescence spots using IDL software. Then, raw intensity data were analyzed using MATLAB script. Those molecules were selected that displayed donor and/or acceptor photobleaching, or showed clear dynamics with anti-correlated donor-acceptor fluorescence for further analysis. The number of photobleaching steps was then used to determine how many donor and/or acceptor dyes were present, with only molecules containing one donor and one acceptor used for further analysis. Intramolecular G4 was designed in such a way that it contains one donor and one acceptor, therefore pre-selection of molecules was not required. In contrast, an intermolecular G4 has many possible combinations of donor and acceptor molecules because of its design, so pre-selection of molecules that contain one donor and one acceptor was necessary. Typically, in a field of view we observed ~29% molecules containing one donor and one acceptor fluorophore showing ~0.6 FRET,~6% molecules showing above 0.75 FRET and the rest of the molecules either had one donor (~50%) or two donor molecules and no acceptors (~15%). The collected FRET traces were further analyzed using the vbFRET algorithm (https://sourceforge.net/projects/vbfret/; *Bronson et al., 2013*) to find possible FRET states and transition frequencies among these FRET states with a Hidden Markov Model (HMM). All graphs were generated and fitted using Igor software.

To measure rate constants for G4 unfolding, dwell time histograms were constructed and fitted to a gamma function of the form:

$$f(t) = (\tau)^{(N-1)}\exp(-k\tau)$$

where $\tau$ is the dwell time, $k$ is the observed rate constant of the transition between FRET states, and $N$ is the number of hidden steps (*Floyd et al., 2010*; *Syed et al., 2014*). Chi-square values represent the goodness of fit of the above curves, and p-values were calculated using the chi-square values and their corresponding degree of freedom (n-1, where n is the number of bins used to construct a particular histogram). SEMs of the rate constants were calculated by dividing the standard deviation of the reciprocal of the mean of the dwell time values ($\tau$) by the square root of the number of molecules in that data set.

## Gaussian fitting of the cumulative FRET histograms

A multiple-Gaussian fit model was applied to FRET histograms generated by binning many FRET trajectories, to obtain mean FRET values. Errors represent the curve-fitting errorstandard deviation of the data.

## Telomerase molecular modeling

The fasta sequence of the human Telomerase Reverse Transcriptase (TERT) was obtained from Uniprot (ID:O14746). Swiss-model (*Waterhouse et al., 2018*) was used to create a homology model using all possible templates. The highest quality model (based upon a combination of the best Q-Mean (−7.33) and Mol Probability (2.08) scores, and the amino acid properties (>82% Ramachandran favoured)) was chosen. This model utilized the cryo-EM structure of *Tetrahymena thermophila* TERT as a template, which has 19.03% sequence identity to human TERT (PDB code: 6D6V *Jiang et al., 2018*).

Using the NMR structure of parallel G4 formed from 5'-TTAGGGT-3' (PDB code: 1np9 *Gavathiotis and Searle, 2003*), the torsion between the T and A nucleotides was altered in a single strand of the G4 in order to flip the bases out, allowing them to interact with RNA. The position of the RNA-DNA hybrid in the *Tetrahymena* cryo-EM structure (*Jiang et al., 2018*) was then used to

position RNA-10C and the flipped G4 in the active site, allowing the DNA 3' end to be located at the catalytic aspartates. In this orientation, the remaining three strands of the G4 overlapped where the CTE is positioned in the cryo-EM structure. The torsions of the beta linker amino acids (Arg938 and Thr939) were altered to enable an extended conformation, removing the DNA-protein clashes. The whole complex was then minimized for 10 ns and 100 ns using Desmond (*Bowers et al., 2006*). Figures were created using PyMOL (The PyMOL Molecular Graphics System, Version 2.3.2 Schrodinger, LLC).

## Acknowledgements

We thank Omesha Perera for plasmid construction, George Lovrecz and Tram Phan for production of telomerase-expressing cells, Timothy Adams for providing pAPEX vectors and for consultation on overexpression of telomerase, and Corinne Getta for preparation of PhenDC3. The work in the Bryan laboratory was supported by Cancer Council NSW project grants RG 11–07 and RG 16–10, a Denise Higgins PhD Scholarship and a Kids Cancer Alliance PhD Top Up Scholarship from Cancer Institute NSW (to ALM). The work in the van Oijen laboratory was supported by an Australian Laureate Fellowship to AMvO (FL140100027). SBC was supported by the Ernest and Piroska Major Foundation. MLB was supported by an Australian Postgraduate Award. Funding in the Damha laboratory was provided by the National Science and Engineering Council of Canada (Discovery NSERC grant). SS was supported by Center of Excellence for Innovation in Chemistry (PERCH-CIC) and Research Unit of Natural Products and Organic Synthesis for Drug Discovery (NPOS 405/2560).

## Additional information

### Funding

| Funder | Grant reference number | Author |
|---|---|---|
| Cancer Council NSW | RG 11-07 | Tracy M Bryan |
| Cancer Institute NSW | | Aaron Lavel Moye |
| Australian Research Council | FL140100027 | Antoine M van Oijen |
| Ernest and Piroska Major Foundation | | Scott B Cohen |
| Natural Sciences and Engineering Research Council of Canada | | Masad J Damha |
| Centre of Excellence for Innovation in Chemistry | PERCH-CIC | Siritron Samosorn |
| Research Unit of Natural Products and Organic Synthesis for Drug Discovery | NPOS 405/2560 | Siritron Samosorn |
| Cancer Council NSW | RG 16-10 | Tracy M Bryan |

The funders had no role in study design, data collection and interpretation, or the decision to submit the work for publication.

### Author contributions

Bishnu P Paudel, Conceptualization, Software, Formal analysis, Investigation, Visualization, Methodology, Writing - original draft, Designed smFRET experiments, performed and analysed smFRET experiments, wrote the manuscript; Aaron Lavel Moye, Conceptualization, Formal analysis, Investigation, Visualization, Writing - review and editing, Designed experiments, purified telomerase and performed ensemble telomerase assays; Hala Abou Assi, Roberto El-Khoury, Resources, Formal analysis, Investigation, Writing - review and editing, Synthesis and purification of modified oligonucleotides; Scott B Cohen, Resources, Investigation, Methodology, Writing - review and editing, Purified telomerase and performed ensemble telomerase assays; Jessica K Holien, Formal analysis, Investigation, Writing - review and editing, Performed molecular modeling; Monica L Birrento,

Resources, Investigation, Writing - review and editing, Characterization of G4 ligands; Siritron Samosorn, Resources, Supervision, Funding acquisition, Methodology, Writing - review and editing, Synthesis and characterization of G4 ligands; Kamthorn Intharapichai, Investigation, Writing - review and editing, Synthesis and characterization of G4 ligands; Christopher G Tomlinson, Supervision, Methodology, Writing - review and editing, Analysis of ensemble telomerase assays; Marie-Paule Teulade-Fichou, Resources, Supervision, Methodology, Writing - review and editing, Synthesis and characterization of G4 ligands; Carlos González, Resources, Supervision, Methodology, Writing - review and editing, Synthesis and purification of modified oligonucleotides; Jennifer L Beck, Resources, Supervision, Funding acquisition, Writing - review and editing; Masad J Damha, Conceptualization, Resources, Supervision, Funding acquisition, Methodology, Writing - review and editing, Synthesis and purification of modified oligonucleotides; Antoine M van Oijen, Conceptualization, Resources, Supervision, Funding acquisition, Methodology, Writing - review and editing, Designed and analysed smFRET experiments, edited the manuscript; Tracy M Bryan, Conceptualization, Formal analysis, Supervision, Funding acquisition, Investigation, Visualization, Methodology, Writing - original draft, Writing - review and editing, Purified telomerase and performed ensemble telomerase assays, analysed data, wrote the manuscript

## Author ORCIDs
Bishnu P Paudel (iD) https://orcid.org/0000-0003-3518-3882
Hala Abou Assi (iD) http://orcid.org/0000-0003-1087-8805
Roberto El-Khoury (iD) https://orcid.org/0000-0002-8068-4566
Jessica K Holien (iD) https://orcid.org/0000-0002-8735-2871
Carlos González (iD) https://orcid.org/0000-0001-8796-1282
Masad J Damha (iD) https://orcid.org/0000-0002-4458-1623
Antoine M van Oijen (iD) https://orcid.org/0000-0002-1794-5161
Tracy M Bryan (iD) https://orcid.org/0000-0002-7990-5501

## Decision letter and Author response
Decision letter https://doi.org/10.7554/eLife.56428.sa1
Author response https://doi.org/10.7554/eLife.56428.sa2

## Additional files

### Supplementary files
• Supplementary file 1. Table of oligonucleotides used in this study.

• Transparent reporting form

### Data availability
All data generated or analysed during this study are included in the manuscript and supporting files. Source data files have been provided for all summary graphs.

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
