## [Decision Letter]

Thank you for submitting your article "A mechanism for the extension and unfolding of parallel G-quadruplexes by human telomerase at single-molecule resolution" for consideration by *eLife*. Your article has been reviewed by three peer reviewers, one of whom is a member of our Board of Reviewing Editors, and the evaluation has been overseen by John Kuriyan as the Senior Editor. The reviewers have opted to remain anonymous.

The reviewers have discussed the reviews with one another and the Reviewing Editor has drafted this decision to help you prepare a revised submission.

Summary:

Telomerase is the enzyme tasked with maintaining telomeres at the ends of linear eukaryotic chromosomes. Telomere DNA sequence is repetitive and guanine rich in most organisms, and the tandem arrays of guanine bases can form multiple planar G-quartet motifs that are stabilized by Hoogsteen hydrogen bonding. In turn, several G-quartets can stack upon each other to form G-quadruplex (GQ) DNA folds, characterized by their very polymorphic structural properties and high melting temperatures. This manuscript by the van Oijen and Bryan groups builds on previous work by the Bryan lab where they showed that purified human telomerase disrupts and processively extends parallel intermolecular G4 structures. In this current manuscript, the authors present a new smFRET-based approach to observe the dynamics and intermediates during telomerase-mediated resolution of parallel G4 structures. Combined with bulk biochemistry, these measurements establish a three-step mechanism for the unfolding and resolution of parallel intermolecular G4 structures.

This manuscript provides significant new mechanistic insight into the function of human telomerase, and the single-molecule data and analyses are overall of excellent quality. A subset of the conclusions seems to be overstated; here the manuscript would benefit from a toned-down interpretation of the data presented (see revisions). Please note that in response to the current COVID-19 situation, only textual changes are required, however already existing data addressing the points raised below should be included, if available.

Essential revisions:

1) The FRET values in the study are interpreted quite loosely. For example, there is no control experiment presented that demonstrates the low FRET state observed in their experiments is in fact a completely resolved GQ. While the interpretation for this experiment is certainly reasonable based upon previous smFRET studies of GQ structures, the authors should ideally substantiate this assignment with mutant DNA substrates or perhaps using Li+ in their buffers. If such new data cannot currently be acquired, the authors should better substantiate their claim that the low FRET state is unfolded by more explicitly referring to other published work. The assignment of all FRET states in this paper would additionally benefit from some structural modeling to estimate expected distances.

2) Related to the first point, it is surprising that both the intra- and intermolecular GQ FRET experiments give essentially the same signal? The interpretation of the two-step drop seems plausible for the intramolecular GQ, but for the intermolecular structure it is unclear why initial docking on one of the four 3' ends would impact the positioning of the FRET dyes located on the 5' ends of the DNA substrate (as drawn in Figure 2A)? Also, is ~0.55 FRET a reasonable value for the schematic drawings shown in Figure 2A, in other words what is the known distance between dyes on neighboring strands of an intermolecular GQ? One would expect this to be quite close (~2-3 nm) and therefore give rise to a much higher initial FRET state. It is also unclear why telomerase binding and remodeling of this tetrameric structure would not cause the structure to fall apart, and therefore give rise to a zero FRET state or loss of signal altogether. Is it possible this is happening and just not being detected in these experiments? The authors should address this in the text.

3) Given the results from Hwang, Myong et al., 2012 (not cited) on the binding and resolution of GQ structure by POT1-TPP1 protein, which occurs in three steps, is it unexpected that two steps is what is observed in the present work? The authors should discuss their results in light of this earlier work.

4) The authors' use of active versus passive unwinding mechanisms for telomerase-mediated GQ unfolding is somewhat unclear. The argument for 'active' unwinding is based upon experiments performed with a short RNA oligo presented in Figure 3 and Supplementary Figure 3, in which this short RNA is meant to mimic the RNA template within the assembled telomerase RNP. The authors state that given the concentration dependence of the RNA-induced FRET drop, the mechanism must therefore be 'active' – but why is this necessarily true? The authors do not show any kinetic data for this experiment, therefore it is difficult to ascertain whether the process displays zero or first-order kinetics with respect to oligo-induced disruption of the GQ structure. Moreover, since there is no energy cofactor (i.e., dNTP) in this experiment, is not the mechanism, by definition, passive? The authors should articulate specifically how they are using the word 'active' in this section of the study? In addition, the authors seem to think the experiments in this figure rule out the possibility of transient intrinsic GQ dynamics being 'captured' by the RNA oligo (or telomerase RNP). However, previous work including that by Green et al., 2003 (not cited) describes a similar experiment with PNA oligos and comes to the opposite conclusion. Also, it seems unlikely that a 10 nt RNA oligo fully captures the properties of the RNA template within the telomerase RNP context – so a more convincing experiment to support the conclusion that it is just the RNA template that is required for the first FRET drop would be to use either a mutant template RNA and/or mutant DNA substrate. At the very least, the authors should discuss the limitations of using a 10 nt RNA oligo to mimic the properties of the telomerase RNA template within the assembled complex.

5) In Figure 4, the authors present very interesting smFRET data for the system being 'walked' through the telomere DNA synthesis reaction through the use of ddNTP terminators. This experiment, and indeed aspects of the study more generally are similar to work from Parks and Stone, 2014 (not cited), in which DNA primer dynamics are directly observed by smFRET in human telomerase as the primer length approaches the template boundary. Ultimately, the authors claim that the experiment in Figure 4I argues that telomerase translocation is required to fully disrupt the GQ structure, evidenced by the second FRET drop upon incorporation of a ddGTP. The conclusion that translocation leads to GQ unfolding is too strong given the data, since in this experiment there is no evidence of translocation having occurred. Typically, translocation would be thought of as being complete once the necessary rearrangements required to allow re-priming for subsequent rounds of telomere DNA synthesis have occurred – but it is not evident this is happening in the experiments described since there is a chain terminator. Therefore, the authors should revise their use of the term translocation so as to avoid an over-interpretation of their very interesting data.

6) In Figures 5 and 6 the effects of GQ binding ligands are studied and the conclusion is that these compounds do not completely inhibit telomerase in the smFRET assay. Why are the results of the gel-based assay for the F-22G3 and the smFRET experiments not more similar? It appears in Figure 5L that NMM and PhenDC3 almost completely inhibit the telomerase activity, whereas SST16 does not have as strong of an effect (the authors should quantify the relative activities in this experiment and include them as part of the figure). In contrast, all three compounds yield similar results in the smFRET assay as shown in the bar plots in Figure 5J. The authors should explain this apparent discrepancy, as this might seem to imply that the two-step FRET drop is not correlating with catalytic activity in this experiment. The authors also comment on the impact of PhenDC3 causing a change in the product profile after four repeats 'most likely resulting from G-quadruplex stabilization within the product DNA'. This result is in close accord with the recent study by Jansson et al., 2019 (not cited) wherein the possibility of GQ folding within the product of human telomerase during catalysis is described. In addition, in the Discussion, the authors relate the thermal stabilization of the structure by these compounds as evidence for 'telomerase not simply exploiting inherent instability in the G4 structure…' However, smFRET experiments by Jena, Ha et al. JACS 2009 (not cited) – showed that compounds that increase the thermal stability of the GQ fold do not necessarily restrict the conformational dynamics of the structure.

7) While the two-step FRET signal observed throughout the study is very interesting, the model put forth in Figure 7 is not very satisfying. Throughout the paper (including in the title) the authors describe how these data lead them to a mechanism for how telomerase acts as a GQ resolvase. However, the authors do not make an effort to discuss their data within a structural context based upon the medium-resolution cryo-EM structure of human telomerase (Nguyen et al., 2018, not cited) or the higher-resolution cryo-EM structure of Tetrahymena telomerase (Jiang et al., 2018, not cited). Why is there no attempt to explain how the proposed GQ unfolding might occur? For example, specifically how steric hindrance by the enzyme would cause unfolding during catalysis, but still allow binding to occur at the start of catalysis? Moreover, in the case of the tetrameric GQ, why would activity cause the structure to disassemble as would be suggested by the cartoon in Figure 7 (see comment above)? The authors should clarify this in the text.

8) The dwell time distribution shown in Figure 1N does not seem to be described particularly well by a single exponential decay. Nonetheless the authors argue that the distribution indicates only one rate-limiting step. Given that the observed reaction is a multi-step process, a more complex kinetic model might be required. If the authors have data recorded at various different dNTP concentrations, they should be included and discussed.

Related to this point, the second FRET decrease observed with different G4 structures, appears to occur abruptly/instantaneously. Can intermediates during the transition be visualized (at higher frame rates or limiting dNTP concentrations)? If the transition times can be quantified, what is the underlying distribution?

9) Have EMSAs or similar experiments been performed to clarify that the RNA is actually binding / invading the G4, whereas nothing happens with DNA? Similarly, it would be informative to see binding data (Kd) for telomerase to the G4 structures used here. Also, is the initial FRET change from 0.5 to 0.3 correlated in time with telomerase binding? In other words, does the time lag from injection to FRET shift match the telomerase on-rate, or does the FRET change require more time (indicating a more complex process)?

10) Is the FRET change observed by telomerase / RNA binding reversible, e.g. if telomerase dissociates? Similarly, given that very high concentrations of RNA were required to observe the effect seen in Figure 3F, RNA is most likely only loosely bound. It would be interesting to see if the G4 reverts when the RNA dissociates.

11) The authors find that G4-stabilizing ligands double the number of G4 that are not opened. This is not really discussed. Could it be that the G4s that are actually unwound are not really bound by the ligands? Does this effect depend on the ligand concentration?

There is also an apparent discrepancy between the amount of G4s that are stabilized, the change in unfolding rate (Figure 5K), and the extension rate (Figure 5L). For example, PhenDC3 results in a minor slowing down of G4 unfolding in Figure 5K but seems to completely inhibit extension (Figure 5L). These discrepancies should be discussed.

---

## [Author Response]

Essential revisions:1) The FRET values in the study are interpreted quite loosely. For example, there is no control experiment presented that demonstrates the low FRET state observed in their experiments is in fact a completely resolved GQ. While the interpretation for this experiment is certainly reasonable based upon previous smFRET studies of GQ structures, the authors should ideally substantiate this assignment with mutant DNA substrates or perhaps using Li+ in their buffers. If such new data cannot currently be acquired, the authors should better substantiate their claim that the low FRET state is unfolded by more explicitly referring to other published work. The assignment of all FRET states in this paper would additionally benefit from some structural modeling to estimate expected distances.

We agree that controls showing the FRET from a denatured or resolved G4 are important. Since we were able to carry out a small amount of experimental work despite coronavirus restrictions, we performed the suggested experiments. After placing each of the two G4 constructs on the surface in potassium-based buffer, we replaced the potassium buffer with Li^+^- buffer and monitored the FRET signal. The data show that both G4s exhibited a drop in FRET signal to a value similar to that observed in the presence of telomerase and dNTPs (see Figure 2—figure supplement 1 and Figure 4—figure supplement 2), supporting the proposition that this FRET value is indicative of an unfolded G4.

The intramolecular G4 (F-22G3) FRET decreased from ∼0.5 to ∼0.14 (new Figure 2—figure supplement 1; subsection “Human telomerase binds, unfolds and extends intramolecular parallel G-quadruplexes”) and a majority of molecules remained at that lowest FRET state during the entire observation window. A minor proportion (35 ± 8%) of molecules showed dynamic transitioning between FRET states but favored the lowest FRET state.

Intermolecular G4 (F-7GGT) also displayed a FRET decrease from ∼0.6 to ∼0.2 after swapping K^+^ for Li^+^ (new Figure 4—figure supplement 2; subsection “Human telomerase also binds, unfolds and extends intermolecular parallel G-quadruplexes”), followed by a loss of FRET altogether, suggesting the G4 structure dissociates from the surface. This behaviour is readily explained by the fact that the biotinylated strand does not bear any fluorophores and that loss of the other strands causes the FRET signal to disappear. The presence of a stable FRET signal at ~0.2 in a majority of molecules prior to disappearance of the FRET signal suggests that the 0.2 FRET state represents an unwound G4, prior to complete diffusion of the non-biotinylated strands. As predicted by one of the reviewers, this behaviour does differ from that in the presence of telomerase + dNTPs (discussed further below), suggesting that the presence of telomerase prevents full dispersion of the unwound strands, and we have addressed this in the aforementioned subsection.

Overall, these results are consistent with our original interpretation whereby G4 extension by telomerase in the presence of dNTPs causes a decrease in FRET due to disruption of the G4 structure.

2) Related to the first point, it is surprising that both the intra- and intermolecular GQ FRET experiments give essentially the same signal?

The similar FRET values between the constructs is probably nothing more than a coincidence. The intra-molecular G4 exhibits FRET centred around 0.5, while inter-molecular has a FRET value centred around 0.6. To better understand the relationship between the expected FRET and the dye positions within the G4, we have now used previously published crystal (intramolecular G4) and NMR structures (intermolecular G4) to estimate the distance between dyes, respectively.

Importantly, we should note that reliable distance estimates obtained from FRET are extremely challenging, with the FRET efficiency depending upon several factors: 1) The length of linkers that are used to attach each fluorescent dye (in our case a C6 linker) will affect dye mobility and increase the distance between dyes, both processes affecting FRET. 2) The quenching of emission due to the presence of Gs in the dye’s vicinity. 3) Solvent polarity affecting FRET transfer (see AO Crockett and CT Wittwer, Anal Biochem [2001] vol 290: 89; S Sindbert et al., 2011; Ishikawa-Ankerhold, Ankerhold and Drummen, 2012).

The above factors notwithstanding, we calculated the FRET efficiency based on an available crystal structure for a version of F-22G3 without the 2'F-araG substitutions (parallel intramolecular GQ, PDB 1kf1). The estimated distance between the two dyes, which are located on the two strands as shown in Author response image 1, is approximately 53 Å including the length of the linkers. Given a Förster radius between AF555 and AF647 of 51 Å, the calculated FRET efficiency for this distance is 0.44, which is close to our experimental FRET value of 0.5. This similarity suggests there is not a significant change in dye emission in this construct due to quenching, with the acceptor dye being located away from the quenching Gs (see Author response image 1).

**Author response image 1. respfig1:** Intramolecular (F-22G3) forming sequence (top) and hybridisation sequence (bottom).

For the intermolecular GQ (F-7GGT), however, a high FRET efficiency is expected as the dyes are attached at the proximal 5’ ends, as pointed out by the reviewers. The estimated distance between dyes is approximately 25 – 35 Å, based on an NMR solution structure of this tetrameric G4 (PDB 1PN9; there is no crystal structure available), together with C-6 carbon linkers for fluorophore attachment. This would yield a theoretical FRET efficiency of ≥ 0.9; however, most of the molecules exhibited FRET efficiency distributed around 0.5 to 0.7 in our experiments. The discrepancy in measured experimental FRET compared to expected FRET suggests that fluorophore quenching occurs in this construct as both members of the FRET pair are close to the G-tetrad plane. Nevertheless, we are confident that the observed ~0.6 FRET state represents a parallel tetrameric G4, since i) there is no other G4 conformation that can form from this short oligonucleotide, and ii) the Li^+^ control above demonstrates a reduction in this signal upon G4 unwinding.

To our knowledge, there are no smFRET studies carried out using intermolecular or intramolecular parallel telomeric G4 constructs to directly compare with our FRET design.

With the caveat that relating FRET values to distances is challenging, we have briefly discussed these points in the revised manuscript (subsections “Human telomerase binds, unfolds and extends intramolecular parallel G-quadruplexes” and “Human telomerase also binds, unfolds and extends intermolecular parallel G-quadruplexes”).

The interpretation of the two-step drop seems plausible for the intramolecular GQ, but for the intermolecular structure it is unclear why initial docking on one of the four 3' ends would impact the positioning of the FRET dyes located on the 5' ends of the DNA substrate (as drawn in Figure 2A)?

We postulate that upon telomerase binding to intermolecular G4, since the template RNA of telomerase needs to hybridise with the 3’ end of one of the four strands of the G4, this will trigger a change in conformation (maybe flipping of the strand) in such a way that two fluorophores move further apart, resulting in a decrease in FRET. We have now performed molecular dynamic simulations of this hybridisation and potential strand flipping, in the context of a homology model of human telomerase (discussed further in response to point 7 below), and we do indeed observe the 5’ ends of the G4 move further apart between 10 ns and 100 ns of the simulation (see Figure 10B).

Also, is ~0.55 FRET a reasonable value for the schematic drawings shown in Figure 2A, in other words what is the known distance between dyes on neighboring strands of an intermolecular GQ? One would expect this to be quite close (~2-3 nm) and therefore give rise to a much higher initial FRET state.

This point is discussed above; the difference between observed and expected FRET strongly suggests the fluorophores are quenched partly by the presence of G-tetrad plane and other experimental conditions.

It is also unclear why telomerase binding and remodeling of this tetrameric structure would not cause the structure to fall apart, and therefore give rise to a zero FRET state or loss of signal altogether.

Yes, we agree with this point. Surprisingly, in our telomerase extension experiments, all molecules remained at a low FRET state before they photobleached. In contrast, when experiments were conducted in the presence of Li^+^- buffer, a complete loss of FRET signal was observed, suggesting complete dissociation of the non-tethered strands from the surface of attachment. This suggests that the presence of telomerase holds the strands of the G4 near each other for a longer period of time, whereas in the absence of telomerase the unwound strands are free to diffuse away. This has been discussed in the revised manuscript (subsection “Human telomerase also binds, unfolds and extends intermolecular parallel G-quadruplexes”).

Is it possible this is happening and just not being detected in these experiments? The authors should address this in the text.

Yes, this is one possibility. However, in our experiments in K^+^-buffer in the presence of telomerase, we observed that the majority of molecules displayed acceptor dye photobleaching followed by photobleaching of the donor dye. If the strands completely dissociated, one would expect the disappearance of both signals at the same time, as occurred in the Li^+^ control. This point has been discussed in the revised manuscript (subsection “Human telomerase also binds, unfolds and extends intermolecular parallel G-quadruplexes”).

3) Given the results from Hwang, Myong et al., 2012 (not cited) on the binding and resolution of GQ structure by POT1-TPP1 protein, which occurs in three steps, is it unexpected that two steps is what is observed in the present work? The authors should discuss their results in light of this earlier work.

Hwang et al., 2012, showed that POT1 unfolds a 24 nt G4 in four steps, whereas the POT1-TPP1 complex caused a 2-step FRET decrease followed by dynamic folding and unfolding. The mechanism of GQ unfolding by POT1 and the POT1-TPP1 complex is very different to that of telomerase; our data suggest that telomerase can directly hybridize to G4 via its RNA template, whereas POT1 unfolds G4 by a conformational selection mechanism in which G4 unfolding occurs prior to POT1 binding (Chaires et al., 2020). The four different steps of FRET reduction in the case of POT1 are attributable to the two OB folds of each of two POT1 molecules binding to the four “arms” of a G-quadruplex sequentially as it unfolds (Hwang et al., 2012). In our case, our model is that the first step of FRET reduction arises due to a conformational change upon RNA hybridization to the 3’ end of the G4, and the second step is due to complete unwinding upon telomerase extension. We have added mention of the POT1 results in Hwang et al., 2012 to the paragraph in the Discussion comparing the unique mechanism of telomerase G4 unwinding with that of other proteins.

4) The authors' use of active versus passive unwinding mechanisms for telomerase-mediated GQ unfolding is somewhat unclear. The argument for 'active' unwinding is based upon experiments performed with a short RNA oligo presented in Figure 3 and Supplementary Figure 3, in which this short RNA is meant to mimic the RNA template within the assembled telomerase RNP. The authors state that given the concentration dependence of the RNA-induced FRET drop, the mechanism must therefore be 'active' – but why is this necessarily true? The authors do not show any kinetic data for this experiment, therefore it is difficult to ascertain whether the process displays zero or first-order kinetics with respect to oligo-induced disruption of the GQ structure.

We used the term “passive” to mean a mechanism in which the enzyme traps an unfolded G4 in an equilibrium population, i.e. binding of the enzyme to the DNA is preceded by spontaneous G-quadruplex unwinding (as in the case of POT1; Chaires et al., 2020). “Active” referred to any mechanism that is NOT passive trapping, involving direct binding of telomerase to the G-quadruplex. However, the reviewer is correct that we did not measure the kinetics of RNA oligo-induced G4 unwinding. Unfortunately, because of the nonlinear nature of flow in the microfluidic flow cells, it is not possible to accurately measure the time between sample injection and the FRET drop. First, the starting times are measured from the time the RNA is injected. This is followed by a significant delay time (probably most of the delay time we see) before it physically enters the flow cell. Because of fluid dynamics, the RNA will not arrive in a coherent manner at the slide surface, but rather will show up as a gradually increasing concentration, making it impossible to measure binding time in an unambiguous manner. Furthermore, the microfluidic device varies from day to day in ways that affect flow rate. Therefore, it is not practical to measure meaningful initial binding kinetics. We are therefore unable to determine whether the concentration dependence of RNA-induced unwinding is first-order with respect to the RNA, so we have removed the sentence “the concentration-dependent unfolding by the RNA oligonucleotide (Supplementary Figure 3B, C) also supports an active invasion mechanism”.

Nevertheless, we believe that the fact that no FRET drop is observed in the presence of the same concentration of a DNA oligo is still strong evidence that RNA-mediated unfolding is NOT due to passive trapping of an unfolded G4, since this would also occur with a DNA oligo.

Moreover, since there is no energy cofactor (i.e., dNTP) in this experiment, is not the mechanism, by definition, passive? The authors should articulate specifically how they are using the word 'active' in this section of the study?

As discussed above, we were using “passive” to indicate a mechanism involving passive trapping of a spontaneously-unwound G4, rather than a mechanism that does not need the energy of nucleotide hydrolysis. There are other examples of G4-unwinding proteins whose mechanism involves neither passive trapping nor ATP hydrolysis. For example, the ability of RPA to unwind G4 is independent of the stability of the G4, and does not need ATP; the mechanism is thought to involve RPA binding to ssDNA loops in the G4 structure (Qureshi et al. J Phys Chem 2012; Ray et al., 2013). Nevertheless, to avoid ambiguity we have more extensively defined what we mean by a “passive” mechanism in the text (Discussion), and we have removed the term “active unwinding”.

In addition, the authors seem to think the experiments in this figure rule out the possibility of transient intrinsic GQ dynamics being 'captured' by the RNA oligo (or telomerase RNP). However, previous work including that by Green et al., 2003 (not cited) describes a similar experiment with PNA oligos and comes to the opposite conclusion.

The study by Green et al., 2003, (and most other previous work) used a different G4 to those in our study. They measured the kinetics of PNA probe binding to an antiparallel telomeric G4 folded in 100 mM NaCl, a quadruplex known to have relatively low stability (*T*_m_ of 47°C in their study, and see Balagurumoorthy and Brahmachari, JBC, 1994). They found that the unfolding was zero order with respect to the PNA oligo, suggesting that the initial step is a rate-determining internal rearrangement of the quadruplex. In contrast, the parallel G4 we used in this study were highly stable and showed NO unwinding over the time course of our smFRET experiments. Furthermore, a DNA oligo did not demonstrate ANY ability to trap the G4 structure within the time of our experiments. Conversely, RNA binding to G4 was observed within a few minutes. If there were transient intrinsic G4 dynamics, a DNA oligo would also have been able to capture the unfolded form. This point is now clarified in the revised manuscript (Discussion).

Also, it seems unlikely that a 10 nt RNA oligo fully captures the properties of the RNA template within the telomerase RNP context – so a more convincing experiment to support the conclusion that it is just the RNA template that is required for the first FRET drop would be to use either a mutant template RNA and/or mutant DNA substrate. At the very least, the authors should discuss the limitations of using a 10 nt RNA oligo to mimic the properties of the telomerase RNA template within the assembled complex.

We fully agree with this point; the telomerase RNP complex would have far more points of contact with the G4 than the RNA oligo, which is likely the reason that a high concentration of oligo was needed to see unwinding. We have not had the opportunity to perform any experiments with mutant version of those oligos, but we have edited the text to make the point that a 10 nt RNA oligo does not fully mimic the properties of the telomerase RNA template within the assembled complex (subsection “The RNA template sequence is involved in partial unfolding of G-quadruplex structure” and Discussion).

5) In Figure 4, the authors present very interesting smFRET data for the system being 'walked' through the telomere DNA synthesis reaction through the use of ddNTP terminators. This experiment, and indeed aspects of the study more generally are similar to work from Parks and Stone, 2014 (not cited), in which DNA primer dynamics are directly observed by smFRET in human telomerase as the primer length approaches the template boundary. Ultimately, the authors claim that the experiment in Figure 4I argues that telomerase translocation is required to fully disrupt the GQ structure, evidenced by the second FRET drop upon incorporation of a ddGTP. The conclusion that translocation leads to GQ unfolding is too strong given the data, since in this experiment there is no evidence of translocation having occurred. Typically, translocation would be thought of as being complete once the necessary rearrangements required to allow re-priming for subsequent rounds of telomere DNA synthesis have occurred – but it is not evident this is happening in the experiments described since there is a chain terminator. Therefore, the authors should revise their use of the term translocation so as to avoid an over-interpretation of their very interesting data.

The data in Parks and Stone, 2014, demonstrate that addition of a ddGTP opposite the end of the telomerase template sequence is followed by realignment of the DNA primer and rehybridization with the upstream region of the template (Figure 5B in their paper). Therefore, chain termination does not necessarily preclude translocation and rehybridization. There is no reason to believe that this part of the translocation process is not happening in our system also, since the G-quadruplex substrate is compatible with processive addition (new Figures 1 and 3 of our manuscript), and we have now referenced Parks and Stone, 2014, as evidence that it is likely the process of DNA realignment during translocation that triggers the FRET change we see (Discussion). However, the reviewer is correct that we don’t have direct evidence for this in our own experiments, so we have revised the language around our conclusions, now stating that the data in Figure 4 (now Figure 6) demonstrate that the FRET drop is triggered by synthesis reaching the end of the template, and is *likely* to result from the subsequent enzyme rearrangements during translocation (subsection “Telomerase translocation leads to complete unfolding of G-quadruplex structure”, Discussion).

6) In Figure 5 and 6 the effects of GQ binding ligands are studied and the conclusion is that these compounds do not completely inhibit telomerase in the smFRET assay. Why are the results of the gel-based assay for the F-22G3 and the smFRET experiments not more similar? It appears in Figure 5L that NMM and PhenDC3 almost completely inhibit the telomerase activity, whereas SST16 does not have as strong of an effect (the authors should quantify the relative activities in this experiment and include them as part of the figure). In contrast, all three compounds yield similar results in the smFRET assay as shown in the bar plots in Figure 5J. The authors should explain this apparent discrepancy, as this might seem to imply that the two-step FRET drop is not correlating with catalytic activity in this experiment.

Quantitation of the activity assays in Figures 5 and 6 has now been included as part of the figures (new Figure 7—figure supplement 5, and Figure 8D). Upon looking at these numbers, we realized that the gel we used to illustrate the effect of NMM on F-22G3 in the original manuscript was not representative of the mean reduction in activity over multiple experiments, most likely due to the NMM causing artefactual retardation of product migration in that particular gel (see old Figure 5L, left panel), so we have swapped this for a more representative gel, without this artefact (new Figure 7—figure supplement 5A). So the reduction in activity with NMM is not as dramatic as it originally appeared; nevertheless, PhenDC3 does still result in a greater effect on activity than on smFRET unwinding. The reason for this is likely that the smFRET data reports only on the first cycle of telomerase-mediated addition of three nucleotides to a single substrate molecule, whereas the ensemble assay is a multiple turnover assay requiring both continued extension of a single substrate, and rebinding to new substrate molecules in solution. Indeed, PhenDC3 affects telomerase in a manner independent of G-quadruplexes, as evidenced by its inhibition of extension of an 18 nt linear telomeric substrate (Figure 7—figure supplement 5 and de Cian et al., 2007) and a completely non-telomeric substrate (de Cian et al., 2007); this effect on telomerase may include preventing it binding to a single-stranded DNA substrate (de Cian et al., 2007), which would affect total product synthesis in a multiple turnover reaction. We have now included discussion of this point in the manuscript (subsection “Ligand stabilization of intramolecular parallel G4 partially inhibits but does not prevent G4 unwinding by telomerase”).

The authors also comment on the impact of PhenDC3 causing a change in the product profile after four repeats 'most likely resulting from G-quadruplex stabilization within the product DNA'. This result is in close accord with the recent study by Jansson et al., 2019 (not cited) wherein the possibility of GQ folding within the product of human telomerase during catalysis is described.

We apologise for missing this citation, which we should have included; we have now done so. This finding has also recently been independently verified (Patrick et al., 2020), so we cited this later paper also (subsection “Ligand stabilization of intermolecular parallel G4 does not inhibit telomerase activity or telomerase-mediated G4 unfolding”).

In addition, in the Discussion, the authors relate the thermal stabilization of the structure by these compounds as evidence for 'telomerase not simply exploiting inherent instability in the G4 structure…' However, smFRET experiments by Jena, Ha et al. JACS 2009 (not cited) – showed that compounds that increase the thermal stability of the GQ fold do not necessarily restrict the conformational dynamics of the structure.

The difference between our study and that of Jena, Ha et al. JACS 2009 is that they used an unmodified human telomeric DNA sequence that adopts a mixture of different G4 topologies that experienced dynamic interconversion between different conformations even in the presence of ligand binding. In our case, smFRET showed no dynamic conversion or unfolding of either of the two G4s we used, with or without ligand binding (compare Figure 2A in their study with Figure 7—figure supplement 3 and Figure 9—figure supplement 1 in our manuscript). Nevertheless, we take the point that thermal stability does not always equal an absence of dynamics, so since we were unable to directly demonstrate a *reduction* in unfolding dynamics in the presence of ligand (because there was no dynamic unfolding detectable in the time span of our assays in the first place), we have modified that sentence to replace the phrase “provides further evidence that telomerase is not simply exploiting inherent instability in the G4 structure to passively extend spontaneously unwound DNA, but is instead actively involved in G4 unwinding” with “provides evidence that thermal stabilization does not provide a barrier to telomerase-mediated unwinding”.

7) While the two-step FRET signal observed throughout the study is very interesting, the model put forth in Figure 7 is not very satisfying. Throughout the paper (including in the title) the authors describe how these data lead them to a mechanism for how telomerase acts as a GQ resolvase. However, the authors do not make an effort to discuss their data within a structural context based upon the medium-resolution cryo-EM structure of human telomerase (Nguyen et al., 2018, not cited) or the higher-resolution cryo-EM structure of Tetrahymena telomerase (Jiang et al., 2018, not cited). Why is there no attempt to explain how the proposed GQ unfolding might occur? For example, specifically how steric hindrance by the enzyme would cause unfolding during catalysis, but still allow binding to occur at the start of catalysis? Moreover, in the case of the tetrameric GQ, why would activity cause the structure to disassemble as would be suggested by the cartoon in Figure 7 (see comment above)? The authors should clarify this in the text.

We thank the reviewers for this helpful suggestion. In response, we have now enlisted the help of a collaborator with expertise in molecular modelling, Dr Jessica Holien from RMIT University in Australia, to help us determine the feasibility of our proposed model in a structural sense.

Dr Holien had previously constructed an hTERT homology model based on the *Tribolium castaneum* TERT crystal structure (Tomlinson et al., 2015). She has now refined this model based on the *Tetrahymena* cryo-EM structure (Jiang et al., 2018). The human cryo-EM structure (Nguyen et al., 2018) was too low a resolution to improve this homology model further, but was used to ensure the resulting model fit in the cryo-EM envelope.

Dr Holien then took the NMR structure of the [7GGT]_4_ quadruplex (Gavathiotis and Searle, 2003), and flipped one strand out to allow it to interact with the 10 nt RNA template. She then used the position of an RNA-DNA hybrid in the *Tetrahymena* cryo-EM structure to position this G4RNA into the active site, with one DNA 3’ end at the catalytic aspartates.

The affinity of telomerase for nucleotides is decreased in the presence of a G4 substrate relative to a linear substrate (Oganesian et al., 2007, Moye et al., 2015), suggesting that the conformation of the active site is perturbed when bound to the bulkier G4. So as we expected, the active site in the modelled enzyme was not large enough to accommodate the G4-RNA complex; it clashed with the C-terminal extension (CTE) of hTERT. However, others have reported that the CTE may pivot around a region called the “beta linker” (Yang and Lee, 2015, Wu, Tam and Collins, 2017; thus, it seemed reasonable that this linker may lengthen to accommodate the quadruplex. The torsions of the beta linker amino acids (Arg938 and Thr939) were therefore altered to enable an extended conformation (shown in Figure 10A with the G4-RNA complex). Note that Figure 10A represents the completely extended form of the CTE; it is likely that during initial binding and extension of the G-quadruplex, the CTE would be less extended than this, and the fully-extended conformation may be needed to allow translocation, as postulated in the papers cited above.

While we emphasize that this model is extremely speculative, we have nevertheless included it in the revised manuscript, since it may help explain our observations. As described above in response to point 2, molecular dynamics simulations suggested that the 5’ ends of the G4 may move slightly apart from each other upon binding in the active site, while the overall structure of the G4 is retained within the boundaries of the active site. Upon DNA-RNA translocation, opening of the active site through pivoting of the CTE domain would allow dissociation of the other 3 DNA strands, resulting in the second step of FRET reduction. We have amended the discussion of our model, and the schematic in Figure 7 (now Figure 10A; described in the subsection “Ligand stabilization of intermolecular parallel G4 does not inhibit telomerase activity or telomerase-mediated G4 unfolding” and in the Discussion), accordingly.

8) The dwell time distribution shown in Figure 1N does not seem to be described particularly well by a single exponential decay. Nonetheless the authors argue that the distribution indicates only one rate-limiting step. Given that the observed reaction is a multi-step process, a more complex kinetic model might be required.

We agree that the dwell time distributions did not fit well with our data and were not very conclusive. According to the reviewer’s suggestion we re-fit our data to a more realistic kinetic model using a gamma function that identifies the number of hidden steps in a complex reaction, if present. For this purpose, we constructed histograms that show distributions of dwell times of inter- and intramolecular G4 (Figure 2—figure supplement 4, Figure 4—figure supplement 4), instead of constructing integrated decay histograms. The new curves show that the data deviate from Gaussian distributions and are more skewed towards the right.

The gamma distribution not only reports on the apparent rate of a complex reaction pathway but also gives the number of rate-limiting steps hidden in the process (Floyd, Harrison and van Oijen, 2010). Both dwell-time histograms were fit to a gamma distribution that gives a number of steps (*N*) greater than two, suggesting that the process consists of at least two rate-limiting steps. Curve fitting was validated by obtaining chi-square values for the fits; the associated p-values all demonstrate no significant difference between expected and observed values (Figure 2—figure supplement 2, Figure 4—figure supplement 4). These new fits have been introduced into the manuscript, together with a discussion of their meaning.

df: degree of freedom

If the authors have data recorded at various different dNTP concentrations, they should be included and discussed.

Yes, we performed telomerase extension experiments at a 100-fold lower dGTP concentration (5 µM). The data reveal that the G4 unfolding rate decreased by 1.8-fold with the same number of steps (*N* = 2.1) in comparison to a higher dGTP (0.5 mM) concentration (new Figure 4—figure supplement 4C). This suggests that the rate of G4 unfolding decreased because the telomerase synthesis process was affected by a change in dGTP concentration. This result is discussed in the revised manuscript (subsection “Human telomerase also binds, unfolds and extends intermolecular parallel G-quadruplexes”).

Related to this point, the second FRET decrease observed with different G4 structures, appears to occur abruptly/instantaneously. Can intermediates during the transition be visualized (at higher frame rates or limiting dNTP concentrations)? If the transition times can be quantified, what is the underlying distribution?

We agree, the transitions seem instantaneous at higher dNTP concentrations. In the experiments at a lower dGTP concentration, we observed that 40 ± 7% molecules of the molecules that changed FRET did indeed display a plateau at 0.4 followed by a gradual drop in FRET from 0.4 to 0.2, compared to a sharp transition at a higher dGTP concentration (new Figure 4—figure supplement 4B). These transitions are gradual and therefore it is challenging to precisely define their endpoints using a change point algorithm. Therefore, we manually analysed those traces to approximate the time it takes to reach 0.2 from 0.4 (which we refer to as τ_unfolding-2_, to distinguish it from the dwell time at 0.4), binning those times into a histogram. The histogram was well fitted with a gamma distribution (Chi-square 4.33 and n = 28) that results in ‘*N*’ greater than 3 with an overall rate constant of 0.44 s^-1^ (new Figure 4—figure supplement 4D). This result suggests that at least three hidden steps are present in the transition from 0.4 to 0.2 FRET states, in addition to the 2-3 steps involved prior to this, and that the kinetics of these steps is affected by dNTP concentration, consistent with a dependence on dNTP incorporation. These data are now added and discussed in the revised manuscript (subsection “Human telomerase also binds, unfolds and extends intermolecular parallel G-quadruplexes”). It should be noted that our data are not sufficient to allow us to assign particular kinetic processes to each of these steps, but we do briefly mention some of the steps that may be involved (e.g. nucleotide binding and incorporation, DNA-RNA repositioning, and an enzyme conformational change to reposition the template back in the active site (Parks and Stone, 2014), Discussion).

9) Have EMSAs or similar experiments been performed to clarify that the RNA is actually binding / invading the G4, whereas nothing happens with DNA? Similarly, it would be informative to see binding data (Kd) for telomerase to the G4 structures used here.

We attempted to observe the RNA oligo binding to the G4 by EMSA analysis, but unfortunately this was not successful; we hypothesize that the RNA is too loosely bound to the G4 for the complex to survive electrophoresis (discussed further in response to point 10 below).

The *K*_m_ for [7GGT]_4_ binding to telomerase was provided in our previous paper (~500 nM; Moye et al., 2015). We have not measured *K*_d_ for either G4, since this information is not relevant in the smFRET experiments, in which the DNA is immobilized to a slide.

Also, is the initial FRET change from 0.5 to 0.3 correlated in time with telomerase binding? In other words, does the time lag from injection to FRET shift match the telomerase on-rate, or does the FRET change require more time (indicating a more complex process)?

There is a significant experimental time lag between injection of telomerase and the FRET change, as discussed above. Due to the nonlinear nature of flow in the microfluidic flow cells and the inability to compare injection times from experiment to experiment, it is not possible to quantitatively interpret the initial time before the first change in FRET.

10) Is the FRET change observed by telomerase / RNA binding reversible, e.g. if telomerase dissociates? Similarly, given that very high concentrations of RNA were required to observe the effect seen in Figure 3F, RNA is most likely only loosely bound. It would be interesting to see if the G4 reverts when the RNA dissociates.

No, telomerase enzyme binding to either G4 is not reversible within our experimental window (3-5 min). No association and dissociation kinetics were observed under our experimental conditions.

However, we did observe dynamic association and dissociation of RNA oligos with G4. We performed experiments at RNA oligo concentrations of 100 μm and 500 uM. In those experiments, we categorised static versus dynamic molecules based on whether a given molecule underwent multiple binding and unbinding events (dynamic) or remained at the binding state once bound (static). We observed that 26 ± 6 and 21 ± 4% molecules show dynamic behaviour in 100 µM and 500 µM RNA concentration, respectively, as shown in the FRET time trajectories (Figure 5—figure supplement 2C and D). As the reviewer predicted, this is indicative that the RNA affinity for G4 is much lower than that of the whole enzyme; we have added these data and discussion of this point to the manuscript (new Figure 5—figure supplement 2, and subsection “The RNA template sequence is involved in partial unfolding of G-quadruplex structure”).

11) The authors find that G4-stabilizing ligands double the number of G4 that are not opened. This is not really discussed.

This is a good point, since this happens in the context of there being no increase in the number of molecules undergoing a single FRET drop. This suggests that the ligands may interfere with telomerase binding to F-22G3, preventing the first FRET decrease that is associated with binding. We have added discussion of this point to the manuscript (subsection “Ligand stabilization of intramolecular parallel G4 partially inhibits but does not prevent G4 unwinding by telomerase”).

Could it be that the G4s that are actually unwound are not really bound by the ligands? Does this effect depend on the ligand concentration?

This is possible, although we consider it unlikely since the degree of thermal stabilization of the population of molecules was so dramatic, particularly for PhenDC3 (see Figure 7—figure supplement 2F). We did not conduct experiments at different concentration of ligands, but we carried out ligand binding at an excess of ligand over DNA. Furthermore, at least in the case of NMM, the rate of unwinding was reduced in the presence of ligand, arguing that most molecules were bound by ligand. Nevertheless, we have added to the text the caveat that we cannot rule out that some of the molecules unwound by telomerase were not bound by the ligands (subsection “Ligand stabilization of intramolecular parallel G4 partially inhibits but does not prevent G4 unwinding by telomerase”).

There is also an apparent discrepancy between the amount of G4s that are stabilized, the change in unfolding rate (Figure 5K), and the extension rate (Figure 5L). For example, PhenDC3 results in a minor slowing down of G4 unfolding in Figure 5K but seems to completely inhibit extension (Figure 5L). These discrepancies should be discussed.

We have now re-fitted the dwell time distribution with a gamma distribution function according to the reviewer’s suggestion under question 8. Overall, the unfolding rates slightly decrease in the presence of ligands, though this was only statistically significant in the case of NMM (Figure K).

The difference between the effects of PhenDC3 in the smFRET analyses and the ensemble activity assays have been discussed in response to point 6 above, and this discussion has been added to the revised manuscript (subsection “Ligand stabilization of intramolecular parallel G4 partially inhibits but does not prevent G4 unwinding by telomerase”).